# SportR: A Benchmark for Multimodal Large Language Model Reasoning in Sports

Haotian Xia[1*] Haonan Ge[3*] Junbo Zou[4*] Hyun Woo Choi[3] Xuebin Zhang[3]
Danny Suradja[3] Botao Rui[3] Ethan Tran[3] Wendy Jin[1] Zhen Ye[5] Xiyang Lin[3]
Christopher Lai[6] Shengjie Zhang[3] Junwen Miao[3] Shichao Chen[3] Rhys Tracy[6]
Vicente Ordonez[1,2] Weining Shen[3] Hanjie Chen[1,2]
[1]Department of Computer Science, Rice University
[2]Ken Kennedy Institute, Rice University
[3]Department of Statistics, University of California, Irvine
[4]College of Sciences, Georgia Institute of Technology
[5]Department of Applied Mathematics and Statistics, Johns Hopkins University
[6]Department of Computer Science, University of California, Santa Barbara
{hx50, hanjie}@rice.edu, weinings@uci.edu

## Abstract

Artificial Intelligence brings powerful new tools to sports, from automated officiating to tactical analysis, but these applications all depend on a core reasoning capability. Deeply understanding sports requires an intricate blend of fine-grained visual perception and rule-based reasoning—a challenge that pushes the limits of current multimodal models. To succeed, models must master three critical capabilities: perceiving nuanced visual details, applying abstract sport rule knowledge, and grounding that knowledge in specific visual evidence. Current sports benchmarks either cover single sports or lack the detailed reasoning chains and precise visual grounding needed to robustly evaluate these core capabilities in a multi-sport context. To address this gap, we introduce SportR, the first multi-sports large-scale benchmark designed to train and evaluate MLLMs on the fundamental reasoning required for sports intelligence. Our benchmark provides a dataset of 4,789 images and 2,052 videos. To enable granular evaluation, we structure our benchmark around a progressive hierarchy of question-answer (QA) pairs designed to probe reasoning at increasing depths—from simple infraction identification to complex penalty prediction. For the most advanced tasks requiring multi-step reasoning, such as determining penalties or explaining tactics, we provide 6,841 high-quality, human-authored Chain-of-Thought (CoT) annotations. In addition, our benchmark incorporates both image and video modalities and provides manual bounding box annotations to test visual grounding in the image part directly. Extensive experiments demonstrate the profound difficulty of our benchmark. State-of-the-art baseline models perform poorly on our most challenging tasks. While training on our data via Supervised Fine-Tuning and Reinforcement Learning improves these scores, they remain relatively low, highlighting a significant gap in current model capabilities. SportR presents a new challenge for the community, providing a critical resource to drive future research in multimodal sports reasoning. The dataset is available at https://github.com/chili-lab/SportR.

## 1 Introduction

The analysis of sports has rapidly evolved from traditional applications in game prediction and statistics (Xia et al., 2022; Oved et al., 2020) into a sophisticated domain for artificial intelligence (Xia et al., 2024a). While the recent advent of Multimodal Large Language Models (MLLMs) (OpenAI, 2024b; AmazonAGI et al., 2025; Gemini Team, 2024; Anthropic, 2024a) has unlocked a new

---

*Equal Contribution.

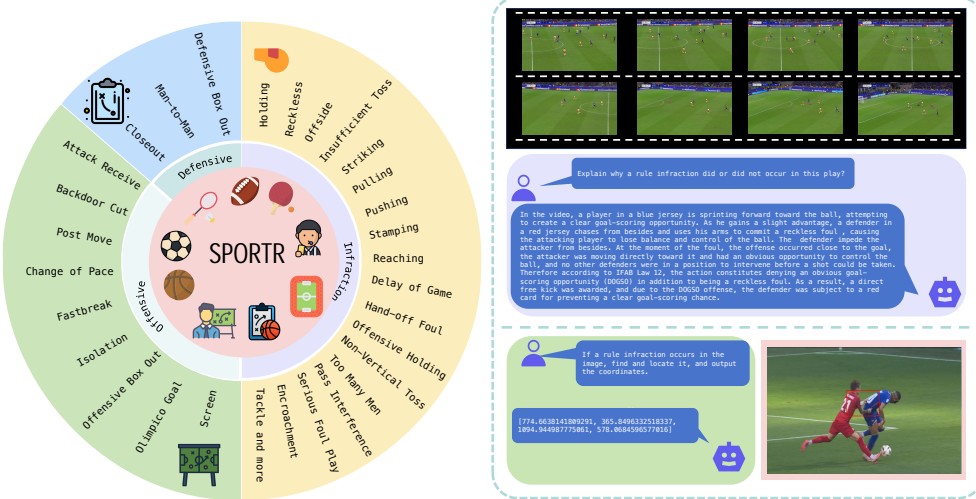

Figure 1: Overview of the SportR benchmark. It consists of two parts, SportsImage and SportsVideo, covering 50 rule infraction categories and 12 different kinds of tactics.

challenge of *sports understanding* by enabling systems to reason about *why* events happen (Gautam et al., 2025; Zhang et al., 2025). The effectiveness of these applications hinges on the capacity of MLLMs for deep rule-based visual reasoning which remains a significant bottleneck. For instance, correctly adjudicating a subtle *hand-check foul* in basketball demands not only recognizing the interaction but also precisely identifying the momentary illegal contact and connecting this visual evidence to sports knowledge.

To address the challenge of complex reasoning in sports understanding, we conceptualize the path to sports intelligence as a pyramid that defines a progressive path for model evaluation. At its base lies perceptual understanding—recognizing players, actions, and basic game states. On tasks at this level, such as identifying the sport being played, current models already achieve high accuracy (Xia et al., 2024b; Chen et al., 2025b), indicating this foundation is primarily established. At the other extreme lie tasks involving the identification of elite professional-level scenarios involving obscure rules or complex tactical sequences. However, the most critical and immediate challenge lies somewhere in the middle: Mastering the identification of fundamental and common fouls and tactics that form the core of sports comprehension for any experienced participant.

Question Answering (QA) has become an effective framework for evaluating MLLM understanding across various domains (Chen et al., 2025a; Yue et al., 2024; Wu et al., 2023; Shao et al., 2024a). However, progress in the sports domain is fundamentally constrained by the limitations of existing benchmarks. While multi-sport benchmarks such as SPORTU (Xia et al., 2024b) provide explanations for evaluation, their annotations are not in the form of fine-grained Chain-of-Thought (CoT) reasoning traces needed to explicitly train the reasoning capabilities of a model. Conversely, specialized benchmarks such as SoccerNet-XFoul (Held et al., 2024) for soccer or FSBench (Gao et al., 2025) for figure skating, offer deep single-sport analysis but do not support the evaluation of multi-sport generalization which is essential for a robust reasoning model. Furthermore, a critical limitation shared across these prior works is the lack of precise visual grounding annotations, making it difficult to verify whether model judgments are tied to specific visual evidence and whether there judgments assess the true source of their reasoning. Studies have shown that while models excel at basic perceptual tasks (e.g., sport identification), they struggle significantly with tasks requiring true comprehension (Xia et al., 2024b; Chen et al., 2025b).

To address these critical gaps, we introduce SportR, a new large-scale, multimodal benchmark designed to train and evaluate fine-grained reasoning for fundamental sports understanding. We focus on a diverse set of five globally popular ball and racket sports – basketball, soccer, table tennis, badminton, and American football – to provide a rich testbed for generalization. Our benchmark is based on a corpus of 4,789 images (from all five sports) and 2,052 videos (from four sports, exclud-

ing badminton), encompassing 50 distinct foul types and 12 fundamental tactics. The cornerstone of our work is a collection of 6,841 high-quality, **fully human-annotated CoT**. We also derive over 20,000 structured question-answer pairs.

Our contributions are threefold:

1. We introduce a novel benchmark for sports reasoning, featuring a progressive QA hierarchy that enables granular evaluation of model capabilities. To provide a gold standard for complex reasoning, we include fully human-annotated CoT traces that contain step-by-step logic for the most challenging tasks.

2. We introduce the first multi-sport benchmark to feature an explicit visual grounding task, requiring models to output the precise bounding box of a rule infraction, directly testing the ability of a model to ground abstract rule knowledge in precise visual evidence.

3. We demonstrate through experiments that SportR is a viable training resource and a relatively challenging benchmark. We show that state-of-the-art training methodologies, including Supervised Fine-Tuning and GRPO-based Reinforcement Learning, have clear performance gains. Also, we observe that these reasoning skills generalize across modalities; a model trained exclusively on our image data exhibits improvement on unseen video tasks. However, even tuned models struggle significantly on the grounding task, only improving from 4.61% to 9.94% on average Intersection over Union (IoU), highlighting the difficulty of our benchmark and the substantial room for future research.

## 2 RELATED WORK

### 2.1 MULTIMODAL SPORTS ANALYSIS

The analysis of sports has rapidly evolved from isolated computer vision (CV) and natural language processing (NLP) tasks—such as action recognition (Shao et al., 2020; Li et al., 2021) or news generation (Huang et al., 2020)—into a domain ripe for the unifying power of MLLMs (Xia et al., 2024a). The advent of foundation models has catalyzed a shift towards systems that can process raw game footage and text to produce nuanced, human-like insights, tackling complex applications from automated commentary (Rao et al., 2024; Baughman et al., 2024) to tactical analysis (Caron & M¨uller, 2023; Zhang et al., 2025). This progress has enabled "sports understanding" applications, where the goal is not merely to describe what is happening, but to reason about why it is happening in the context of rules (Held et al., 2024), strategies, and player interactions (Gautam et al., 2025; Rao et al., 2025). As models demonstrate increasing proficiency in generating fluent and context-aware outputs, the community's focus has shifted towards evaluating and improving their capacity for deep, multi-step reasoning, which is essential for trustworthy and reliable sports analysis. Current MLLMs, such as Gemini 1.5 Pro (Gemini Team, 2024), Claude-3.5-Sonnet (Anthropic, 2024b), and GPT-4o (OpenAI, 2024a) have demonstrated sports understanding abilities. Subsequent research has continued to push the boundaries. For instance, the recent work has employed a new strategy and framework to achieve new state-of-the-art performance on benchmarks (Chen et al., 2025b). However, even with these advances, the accuracy on the most difficult, rule-based tasks remains modest, suggesting that the core challenge of connecting visual perception to abstract rules is far from solved.

This necessitates the development of more challenging and fine-grained benchmarks designed to probe the limits of their performance and reasoning capabilities.

### 2.2 MULTIMODAL SPORTS QA BENCHMARKS

Question answering (QA) has become the standard way for evaluating the reasoning abilities of MLLMs across different domains Chen et al. (2025a); Liu et al.; Yue et al. (2024); Wu et al. (2023); Shao et al. (2024a). In the sports domain, early multimodal QA benchmarks primarily focused on assessing models' ability to recognize actions and answer descriptive or temporal questions. For instance, ActionAtlas (Salehi et al., 2024) which focuses on action recognition via multiple-choice, or Ego-Exo4D (Grauman et al., 2024) which evaluates skill proficiency and pose estimation and Sports-QA (Li et al., 2024c), which leveraged existing action recognition datasets to create a large-

scale VQA benchmark, but its questions do not center on the complex, rule-based scenarios that define competitive play.

More recent benchmarks have begun to address this gap by focusing on deeper forms of reasoning. Notably, SPORTU (Xia et al., 2024b) introduced a comprehensive evaluation benchmark with a multi-level difficulty design, explicitly including "hard" questions that require understanding game rules and tactics. However, while SPORTU provides explanations for evaluation, the limitations are that there are not enough of this type of question, and these rationales were not designed as detailed and fine-grained, human-annotated reasoning processes suitable for training models to perform explicit, step-by-step reasoning. Furthermore, it lacks precise annotations for visual grounding, making it difficult to assess whether a model's judgment is based on specific visual evidence, such as the subtle point of contact defining a foul. In addition, SPORTU primarily relies on multiple-choice questions and slow-motion replays, limiting its utility for training deep reasoning and evaluating true sports understanding. Recent studies suggest that the MCQ format may not accurately reflect an LLM's true capabilities due to misalignment with real-world generation tasks and sensitivity to option ordering (Li et al., 2024d). A detailed comparison is provided in Appendix A.

There are also more recent benchmarks that appear to cover a single sport. For example, SoccerNet-XFoul (Held et al., 2024) provides expert-annotated explanations for fouls, FSBench (Gao et al., 2025) delves into the fine-grained, artistic scoring criteria of figure skating, FineBadminton introduces a benchmark for badminton understanding (He et al., 2025), and SoccerBench (Rao et al., 2025) offers thirteen distinct tasks for multifaceted soccer video analysis. These video-centric datasets focus on a single sport, limiting cross-domain evaluation. While these datasets are invaluable, a critical gap remains: the lack of a multi-sport benchmark that provides rich, human-written reasoning process annotations specifically designed to teach models how to reason, combined with the precise grounding annotations needed to verify that this reasoning is tied to the correct visual evidence.

## 3 SPORTR BENCHMARK

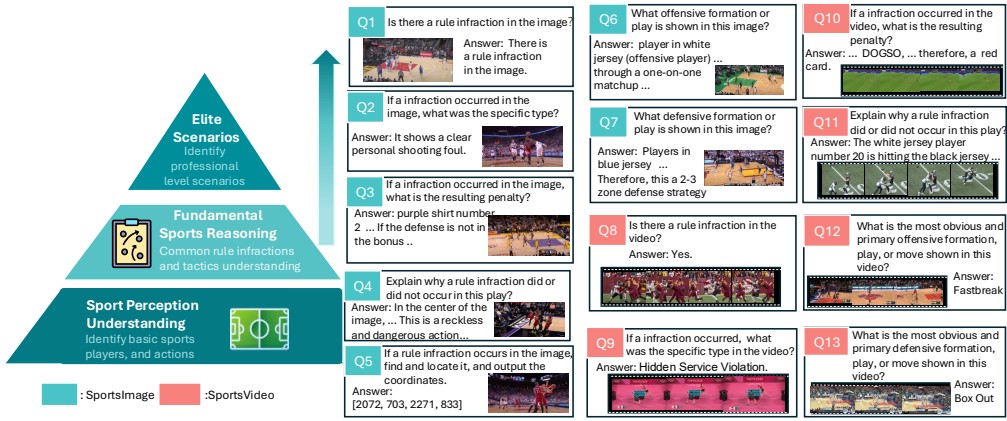

Figure 2: SportR overview. Left: A three-level pyramid frames our evaluation scope—perception (base and well established), fundamental sports reasoning (our focus), and elite scenarios (out of scope). Right: We instantiate a 13-question hierarchy with concrete examples: Q1–Q7 (SportsImage) cover infraction detection, type, penalty reasoning, explanation, grounding by box coordinates, and offensive/defensive tactics; Q8–Q13 (SportsVideo) mirror these tasks in the temporal domain.

To address the limitations in current sports benchmarks, we introduce SportR, a new large-scale, multimodal benchmark designed to train and evaluate the fine-grained visual reasoning capabilities of MLLMs. The cornerstone of our benchmark is a collection of high-quality, human-authoredCoT rationales, each providing a detailed, step-by-step explanation for a fundamental foul or tactical scenario. Our work is guided by a conceptual pyramid of sports understanding that defines a progressive path for model evaluation (Figure 2).

- At the base of this pyramid lies perceptual understanding—recognizing players, actions, and basic game states. On tasks at this level, such as the easy questions in SPORTU, state-of-the-art models and recent work already achieve near-perfect accuracy (Xia et al., 2024b; Chen et al., 2025b), indicating this foundation is largely established.

- At the apex are elite, professional-level scenarios involving highly complex tactical combinations or obscure, controversial edge-case rulings. While a benchmark at this level represents an ultimate goal for the community, it requires a degree of annotation expertise and model capability that is currently beyond reach.

- SportR is therefore positioned to bridge this critical gap, focusing on the essential middle layer: the fundamental fouls and tactics that form the core of deep sports comprehension for any experienced participant. This is a domain where, as we will demonstrate in our experiments, even the most advanced MLLMs struggle profoundly, with grounding performance often failing to surpass a low threshold. By establishing that models must first master these fundamentals before aspiring to elite-level reasoning, we frame SportR as a necessary and foundational next step for the community.

We first designed a suite of QA pairs to probe a model's reasoning at increasing levels of depth. This hierarchy ranges from foundational tasks like infraction identification (Is there a foul?) and classification (What type of foul?), to more complex challenges like penalty prediction (What is the penalty?) and culminating in a precise visual grounding task. To provide a high-quality ground truth for this hierarchy, especially for penalty prediction and tectics recognition tasks, each image or video is annotated with a fully human-annotated CoT.

The benchmark consists of two complementary components:

**SportsImage** – a dataset of 4,789 images designed to test a model's ability to connect abstract rules to precise evidence in static scenes. Each image is paired with a CoT rationale focused on a specific foul or tactical play. This component's progressive QA suite culminates in a unique visual grounding task, where the model is required to output the exact bounding box coordinates of the rule infraction.

**SportsVideo** – a dataset of 2,052 video clips created to extend the reasoning challenge to the temporal domain. This component addresses scenarios where understanding context is only possible by observing motion over time, such as a "traveling" violation in basketball. Each video is also annotated with a comprehensive CoT, from which a similar suite of progressive QA pairs. This part aims to provide a more thorough evaluation benchmark in sports understanding.

Overall, SportR contains 6,841 unique, human-authored CoT rationales, from which we derive over 20,000 structured question-answer pairs. This benchmark fills a critical gap by providing the resources to train models not just to answer what happened, but to reason about why it happened based on explicit, grounded visual evidence.

Our design prioritizes not only assessing a model's reasoning but also verifying that this reasoning is matched to specific visual evidence. To this end, we introduce a novel explicit grounding task, where models must output the precise bounding box coordinates of a rule infraction. We focus this coordinate-based evaluation on SportsImage because fouls are often defined by discrete, localized events—such as a specific point of physical contact—that can be unambiguously annotated and evaluated in a static frame. This provides a clean and tractable setting to evaluate a skill that, as we later demonstrate, poses a significant challenge even for state-of-the-art models. Establishing a reliable baseline for this explicit form of grounding is a critical first step before tackling the greater spatio-temporal complexity of localization in video, which remains an important direction for future work. Examples of our SportsImage and SportsVideo can be found in Appendix F

## 3.1 QUALITY CONTROL

To ensure the highest standard of data quality, our benchmark is **fully human-authored and annotated**. Recognizing that state-of-the-art MLLMs still exhibit significant weaknesses in complex sports reasoning and grounding, as demonstrated by prior benchmarks (Xia et al., 2024b), we decided to discard any model-assisted generation for our annotations. **We posit that creating a reliable and expert-driven ground truth is a critical, pioneering step for a domain where models are known to struggle**.

All CoT reasons and annotations were therefore created by our team of 16 experts, all of whom are authors of this paper. This team includes two former NCAA Division I student-athletes with over 12 years of competitive experience, and 14 other members with at least three years of dedicated training and engagement in their respective sports. This deep, practical expertise was foundational to maintaining the high standard of the benchmark.

A key principle of our data curation was to focus on fundamental and widely understood fouls and tactics, such as a basketball blocking foul or a standard post move, rather than highly specialized or controversial professional-level edge cases. This focus ensures the scenarios, while challenging for AI systems, are grounded in the established, practical expertise of our annotation team. This allows the creation of clear, consistent, and authoritative rationales, minimizing ambiguity and subjectivity.

The process began with a training phase where all annotators collaboratively developed and standardized the guidelines for writing CoT rationales and annotating grounding boxes. Specifically, annotators were trained to follow a strict "Macro-to-Micro" logical flow. The reasoning process consists of three sequential steps: (1) Identify the specific court area and involved parties; (2) Describe the action details and dynamics leading up to the event; and (3) Pinpoint the precise point of contact or critical visual evidence. Each member then worked on small batches of examples, with the group reviewing the outputs to ensure a unified standard of quality and detail before commencing the full-scale annotation.

Following the initial annotation, we implemented a strict verification and filtering protocol. First, each annotator performed a self-review of their work for accuracy. Critically, any instance where an annotator felt uncertain about the correct interpretation or the precise grounding location was flagged and passed to a second expert annotator for an independent review. If a consensus could still not be reached, or if both annotators remained unsure, the data point was discarded from the benchmark to minimize the risk of mislabeling.

## 3.2 SportsImage: Multimodal Image QA

SportsImage is the first component of our benchmark, designed to rigorously evaluate an MLLM's ability to perform fine-grained reasoning and rule-based visual grounding in static scenes. This part covers five sports: basketball, soccer, table tennis, badminton, and American football. It comprises 4,789 images, each capturing a critical moment of gameplay across multiple sports.

Each image is paired with a comprehensive, human-authored CoT rationale. This rationale serves as the ground truth to generate a progressive QA suite of up to seven distinct questions, resulting in over 17,000 structured question-answer pairs for this component alone. These questions are structured to probe a model's reasoning at increasing levels of depth, covering: (Q1) infraction identification, (Q2) foul classification, (Q3) penalty prediction, (Q4) free-form explanation, (Q6) offensive tactic identification, (Q7) defensive tactic identification, and culminating in (Q5) an explicit visual grounding task requiring the model to output precise bounding box coordinates.

**Explicit Visual Grounding:** This is the most challenging and unique task in our QA suite. This final question moves beyond conceptual understanding and requires the model to output the precise bounding box coordinates of the rule infraction. This novel task serves as a test of whether a model's reasoning is connected to specific, localized visual evidence. We focus this coordinate-based evaluation on static images because fouls are often defined by discrete events—such as a specific point of physical contact—that can be unambiguously annotated, providing a clean and high-precision ground truth for this difficult skill. **To the best of our knowledge, SportR is the first benchmark in the sports understanding domain to introduce such a task, directly evaluating a model's ability to ground abstract rule knowledge in precise spatial evidence in sports.**

**Dataset Construction** Our annotation process was guided by our progressive QA hierarchy, with experts providing direct ground-truth answers for each question. We annotated a detailed CoT reasoning process for specific tasks that demand more than a simple answer. For the penalty prediction task (Q3), which requires a multi-step reasoning process, the CoT provides a gold-standard reasoning path from visual evidence to rule application and final judgment. For the explanatory tactic identification tasks (Q6 and Q7), we also offered a human-annotated CoT to illustrate the tactical formation or strategy.

A key feature of our design is that this same expert-written CoT also serves as the definitive ground truth for the free-form explanation task (Q4). This approach allows the CoT to be leveraged flexibly: it can be used as a target for evaluating explanation generation, or as a chain-of-thought prompt to train or guide a model's reasoning process for the more complex tasks.

For the final visual grounding task (Q5), annotators manually drew the tightest possible bounding box around the critical visual evidence—for instance, the exact point of illegal contact between two players' arms. This focus on static images allows for the creation of an unambiguous, high-precision ground truth for localization, ensuring that SportsImage provides a robust and multifaceted benchmark for assessing a model's sports understanding.

### 3.3  SPORTSVIDEO: MULTIMODAL VIDEO QA

SportsVideo is the second component of our benchmark, designed to extend the fine-grained reasoning challenge into the temporal domain. Many fundamental fouls and tactics are inherently dynamic and can only be understood by observing the sequence and context of motion over time. For example, a "traveling" violation in basketball is impossible to judge from a single static frame. SportsVideo complements SportsImage by providing these crucial temporal scenarios, offering a more comprehensive evaluation of sports understanding.

The dataset consists of 2,052 video clips, from which we derive over 6,000 structured question-answer pairs. Like its image-based counterpart, each video is annotated with a comprehensive, human-authored CoT rationale that explains the core foul or tactic. This CoT then serves as the ground truth for a progressive QA suite designed to evaluate spatio-temporal reasoning.

**Dataset Construction** The construction process for SportsVideo is similar to that of SportsImage, adapting our progressive QA hierarchy to the temporal domain. Our annotation team collected short video clips that illustrate fundamental rule infractions or tactics requiring temporal understanding. Following the same protocol, experts provided direct, ground-truth answers for each question in the suite. For tasks demanding a detailed explanation of the sequence of events and reasoning—such as penalty prediction and free-form explanation—annotators authored a comprehensive CoT rationale.

Unlike SportsImage, this suite does not include the explicit, coordinate-based grounding task. The challenge of defining and consistently annotating precise spatial coordinates across multiple dynamic frames is a substantial research problem in its own right. While this is an important avenue for future research, our focus in this component is to establish a strong baseline for a model's ability to perform complex temporal reasoning based on the provided CoT.

## 4  EXPERIMENT

To validate the challenge posed by SportR and to demonstrate its utility as a training resource, our experiments are designed with two primary objectives. First, we establish a comprehensive baseline by evaluating a wide range of state-of-the-art MLLMs in a zero-shot setting. This demonstrates the difficulty of our benchmark for existing models. Second, we investigate the benchmark's effectiveness for model improvement by performing supervised fine-tuning (SFT) and reinforcement learning (RL) on a powerful open-source model. This serves as a proof of concept that our dataset and its human-authored CoT reasons can be effectively used to enhance model capabilities in fine-grained sports reasoning.

### 4.1  BASELINE MODELS

We evaluate a diverse set of leading MLLMs to establish a robust performance baseline on SportR. Our evaluation includes both proprietary and open-source models.

Proprietary Models: We evaluate the latest and most powerful closed-source models, including GPT-5 (a20, 2025), Claude 4.0 (Anthropic, 2025), and Gemini 2.5 Pro (Anthropic, 2025). Access to these models was facilitated through their official APIs. Open-Source Models: We selected a broad range of recently released, high-performing open-source MLLMs, including models from the LLaVA family (LLaVA-OneVision 7B, LLaVA-Next) (Li et al., 2024a;b), the Qwen family (QwenVL-2.5 7B, 72B) (Bai et al., 2025), Deepseek-VL (Wu et al., 2024), and Glm-4.5V (Team et al., 2025).

All baseline evaluations are conducted in a zero-shot setting. For each question in our test set, the model is provided with the image or video and the question text. We use a consistent prompt template across all models and set the temperature as 0.7.

### 4.1.1 MODEL TRAINING ON SPORTR

Beyond zero-shot baseline evaluations, we fine-tuned an open-source model to validate SportR as an effective training resource. We employed a two-stage training process on the Qwen-2.5VL-7B model: an initial Supervised Fine-Tuning (SFT) phase followed by Reinforcement Learning (RL) using the Group Relative Policy Optimization (GRPO) algorithm (Shao et al., 2024b). The training was conducted exclusively on our SportsImage component to demonstrate its value for teaching fine-grained visual reasoning. Full details on our data preparation, SFT/RL methodology, and reward function design are deferred to Appendix B.

### 4.2 EVALUATION

Our evaluation is conducted on the SportsImage test set and the entire SportsVideo dataset, the latter of which is for zero-shot evaluation and cross-modal generalization testing.

We employ two primary metrics: Intersection over Union (IoU) (Everingham et al., 2010) for the visual grounding task (Q5) and an LLM-as-Judge framework for all other text-based QA tasks (detaile prompts are in Appendix C. To mitigate the known issue of self-preference bias where models may unfairly favor their own outputs (Zheng et al., 2023), we use three proprietary models as judges: GPT-5 (a20, 2025), Gemini 2.5 Pro (Comanici et al., 2025), and Claude 4.0 Sonnet (Anthropic, 2025) and report the average score. To validate the reliability of this metric, we conducted a human verification study on a stratified subset of the test set (660 samples) using the same instruction as we provided to LLMs. We observed a strong correlation between the LLM average score and expert human judgments (Pearson $r > 0.65$ across modalities), confirming that our "Average Score" approach aligns well with human evaluation. The correlation map are shown in Appendix E For each generated response, we report the scores from all three judge models as well as the average score to provide a comprehensive and robust assessment of performance.

## 5 RESULT

Our experiments are designed to demonstrate both the profound challenge posed by our benchmark and its utility as a training resource. We present results for the SportsImage and SportsVideo components separately to provide analysis of the model's capabilities in both static and video settings.

### 5.1 PERFORMANCE ON SPORTSIMAGE

We present the performance of all models on the test set of our **SportsImage** component in Table 1. The results are broken down by each of the seven progressive question types, providing a granular view of model capabilities. The evaluation clearly demonstrates both the profound challenge posed by our benchmark and its utility as a training resource.

**Baseline Performance.** The zero-shot results underscore the difficulty of our benchmark. Even the most powerful proprietary model, GPT-5, struggles to achieve high accuracy on the core reasoning tasks. Across all baselines, performance is particularly low on tasks requiring deep, specialized knowledge, such as Foul Classification (Q2) and, most notably, the Explicit Visual Grounding task (Q5), where IoU scores are consistently below 7%. This confirms that existing models lack the sports fine-grained perception and rule-based reasoning abilities that SportR is designed to measure.

**Effect of Training on SportsImage.** After the SFT phase, we observe a dramatic improvement across nearly all tasks. For instance, Foul Classification (Q2) accuracy leaps from 14.4% to 50.7%, validating that our human-authored data provides a strong learning signal. The subsequent Reinforcement Learning (SFT+RL) phase further pushes performance on most tasks, achieving the highest scores on 5 out of 7 categories.

Table 1: Performance comparison across models (v%) on SportsImage. Q1: Infraction identification; Q2: Foul Classification; Q3: penalty prediction; Q4: Free-form Explanation. Q5: Visual Grounding (IoU); Q6: Offensive Tactic Identification; Q7: Defensive Tactic Identification.

| Model | Q1 | Q2 | Q3 | Q4 | Q5 | Q6 | Q7 |
|---|---|---|---|---|---|---|---|
| GPT-5 | 69.19 | 44.21 | 44.49 | **41.34** | 5.70 | **65.75** | 58.82 |
| Claude 4 Sonnet | 52.49 | 26.21 | 30.55 | 14.02 | 3.67 | 31.16 | 73.30 |
| Qwen-2.5VL-72B | 49.97 | 22.92 | 32.61 | 14.64 | 6.93 | 18.83 | 66.96 |
| Llava-Next-8B | 53.26 | 12.01 | 17.85 | 16.81 | 0.00 | 40.04 | 66.99 |
| MiniCPM-V-4.5 | 49.97 | 17.57 | 27.36 | 11.19 | 4.13 | 37.86 | 57.09 |
| Gemini 2.5 Pro | 58.79 | 17.54 | 19.54 | 35.16 | 3.67 | 21.12 | 45.23 |
| DeepSeek-vl2 | 47.56 | 23.09 | 30.41 | 21.19 | 0.91 | 32.04 | 31.70 |
| GLM-4.5v | 51.71 | 20.69 | 26.15 | 20.78 | 4.46 | 26.53 | 27.03 |
| Qwen-VL-7B (Base) | 48.29 | 14.43 | 21.69 | 12.32 | 4.61 | 24.66 | 21.81 |
| Qwen-VL-7B (SFT) | 69.82 | 50.71 | 33.13 | 32.94 | 2.88 | 55.08 | 76.56 |
| Qwen-VL-7B (SFT+RL) | **84.19** | **51.54** | **52.34** | 27.44 | **9.94** | 60.89 | **87.07** |

## 5.2 Performance on SportsVideo

Table 2: Performance comparison across models (%) on SportsVideo. Q8: Video Infraction identification; Q9: Video Foul Classification; Q10: Video Penalty Prediction; Q11: Video Free-form Explanation. Q12: Offensive Tactic Identification; Q13: Defensive Tactic Identification.

| Model | Q8 | Q9 | Q10 | Q11 | Q12 | Q13 |
|---|---|---|---|---|---|---|
| Qwen-2.5VL-72B | 30.19 | 13.84 | 15.90 | 8.67 | 31.86 | 13.10 |
| GLM-4.5v | 24.82 | 17.61 | 19.22 | 12.86 | 19.01 | 9.82 |
| Video-R1-7B | 25.15 | 14.56 | 11.54 | 8.14 | 30.60 | 3.75 |
| MiniCPM-V-4.5 | 27.33 | 2.11 | 2.93 | 5.00 | 35.84 | 7.44 |
| Qwen2.5-VL-7B (Base) | 25.49 | 15.06 | 11.63 | 8.27 | 33.41 | 3.64 |
| Qwen2.5-VL-7B (SFT) | **70.65** | 17.20 | 12.08 | 17.76 | 8.04 | 12.13 |
| Qwen2.5-VL-7B (SFT+RL) | 59.52 | 17.53 | 19.71 | 14.89 | 9.37 | 12.88 |
| Claude 4 Sonnet | 36.93 | 21.37 | 16.99 | 8.15 | 38.08 | 8.25 |
| Gemini 2.5 Pro | 64.93 | 25.69 | 26.71 | 17.81 | 44.18 | **17.87** |
| GPT-5 | 59.17 | **34.39** | **41.83** | **24.02** | **60.82** | 8.42 |

The performance on our SportsVideo part is presented in Table 2. The video tasks show a greater challenge than their image-based counterparts, with overall model performance being lower. As with SportsImage, the SOTA proprietary model, GPT-5, achieves the highest scores on the majority of tasks (Q9-Q12) with a relatively low score, underscoring the difficulty of our benchmark. Another finding of our work emerges from our Qwen-2.5VL-7B (SFT+RL) model, which was trained exclusively on SportsImage data. Despite having no exposure to videos during training, the model demonstrates a promising degree of cross-modal generalization. Its performance on Video Infraction Identification (Q8) improved dramatically from 25.49% to 59.52%, surpassing all other models, including GPT-5. We also observe performance gains over most tasks, except for Q12. This finding underscores the value of SportsImage as a foundational resource for building generalized sports reasoning capabilities.

## 5.3 Error Analysis (New Section added based on reviewer feedback)

To understand the specific challenges posed by SportR, we conducted a manual error analysis on 1,500 failure cases. We sampled 150 images and 150 videos from the test set for each of the six representative models. Errors were classified into five categories: (1) Visual Hallucination, fabricating non-existent objects or events; (2) Domain Knowledge Gap, misapplying sports rules or failing to recognize a standard penalty; (3) Reasoning Error, flawed logical derivation despite correct perception; (4) Format Violation, failing to follow output schema or refusing to answer; and (5) Visual Perception Error, missing critical visual evidence present in the input. The aggregate error distribution is illustrated in Figure 3. For a detailed breakdown by modality (Image vs. Video), please refer to Appendix D.

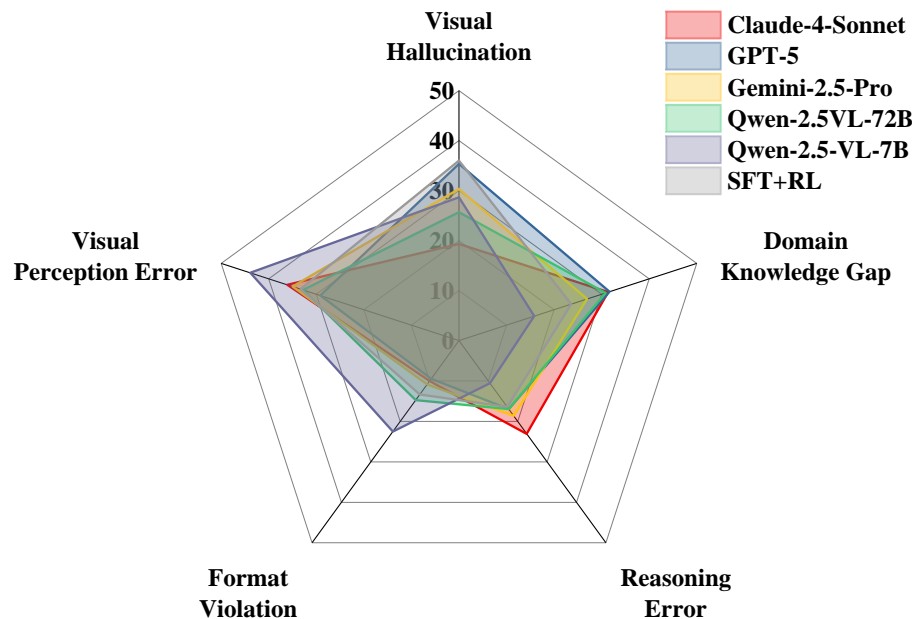

Figure 3: Error type distribution across different MLLMs on overall SPORTR tasks. The analysis reveals that Visual Perception Error is the most common issue, followed by Hallucination Error. Each error type highlights specific model limitations in comprehending the task.

We found in video tasks, errors are overwhelmingly dominated by Visual Perception Error and Visual Hallucination. Across all models, these two categories combined frequently exceed 60-70% of total errors. This indicates that the primary barrier in video is the inability to parse fine-grained, temporal dynamics. Consequently, models rarely reach the stage of successful rule adjudication, resulting in an artificially low "Domain Knowledge Gap."

In contrast, static image tasks—where temporal ambiguity is removed and visual perception is inherently easier—reveal the true extent of the models' limitations in sports logic. We observe a sharp spike in Domain Knowledge Gaps when shifting from video to image. This trend is particularly evident in proprietary SOTA models. For instance, GPT-5's Domain Knowledge Gap rises from 20.00% in video to 36.00% in images. This confirms that the decrease in visual perception errors, expected due to the static nature of images in sports, effectively unmasks the underlying deficiency in reasoning: even when SOTA models successfully perceive the visual evidence, they struggle significantly to map that evidence to the correct abstract sports rule.

## 6    CONCLUSION

In this paper, we introduced SportR, a large-scale benchmark designed to train and evaluate the fine-grained, rule-based visual reasoning of Multimodal Large Language Models. Built upon a foundation of over 6,500 fully human-authored Chain-of-Thought rationales, we aim to provide a benchmark that can be used both for training and evaluation. By integrating tasks across both images and videos, our benchmark provides a holistic assessment of a model's ability in sports understanding. Our experiments demonstrate that while our dataset is an effective training resource, it presents a profound challenge to current models. For example, even after fine-tuning and reinforcement learning, performance on the explicit grounding task remains modest, underscoring the difficulty of connecting abstract rules to precise visual evidence. Crucially, our error analysis reveals that current MLLMs suffer from a fundamental shortage in fine-grained visual perception for dynamic events and a significant gap in aligning visual evidence with abstract domain knowledge. We hope SportR will serve as a critical tool for the community, inspiring advancements in MLLMs and contributing to more robust and reliable real-world sports understanding.

## 7 ACKNOWLEDGMENTS

This project is supported in part by the Rice Ken Kennedy Institute Research Award #1081025 and the U.S. National Science Foundation under Grant No. 2429680. We thank the anonymous reviewers for their thoughtful feedback and constructive suggestions, which helped improve this work.

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

## THE USE OF LARGE LANGUAGE MODELS (LLMs)

During the preparation of this manuscript, we utilized large language models (LLMs) to assist with grammar correction and improve the clarity of the writing.

## A  DETAILED COMPARISON WITH EXISTING BENCHMARKS

While benchmarks like SPORTU (Xia et al., 2024b) pioneered multi-sport evaluation, SportR represents a structural evolution designed to support **training** and **fine-grained reasoning**. The key distinctions are summarized below:

**1. Training vs. Evaluation Utility.** SPORTU's "hard" tasks rely heavily on **Multiple Choice Questions (MCQs)**. While effective for low-cost evaluation, MCQs allow models to guess answers and cannot be used to train generative reasoning capabilities. In contrast, SportR provides over 6,500 **human-authored Chain-of-Thought (CoT)** annotations. This transforms the benchmark from a pure evaluation set into a rich training resource that explicitly teaches models *how* to reason step-by-step.

**2. Visual Fidelity (Normal vs. Slow Motion).** A critical limitation of SPORTU's video subset is its reliance on **slow-motion replays**, which simplifies temporal perception. SportR utilizes **normal-speed videos**, presenting a significantly more realistic challenge. Models must capture fleeting visual cues without temporal assistance, testing true temporal understanding.

**3. Task Scope: Adjudication vs. Perception.** Prior benchmarks often include basic perception tasks (e.g., counting players, identifying colors). SportR filters these out to focus exclusively on the **"Reasoning Gap"**: complex rule-based adjudication and visual grounding. This ensures the benchmark measures deep sports understanding rather than basic visual recognition.

# B   DETAILED MODEL TRAINING METHODOLOGY

Here we provide a comprehensive description of the training process used to validate SportR as a training resource.

## B.1   MODEL TRAINING ON SPORTR

Beyond zero-shot evaluation, we investigate whether SportR can serve as an effective resource for model improvement through a two-stage training process: supervised fine-tuning followed by reinforcement learning. We select Qwen2.5-VL (7B) as our backbone model. The training process exclusively uses the SportsImage component, with the data images partitioned into an SFT set, an RL set, and a test set based on Image ID.

## B.2   TRAINING DATA PREPARATION

For questions, such as infraction identification and foul classification, the target output for the SFT phase is the concise, ground-truth answer enclosed in ¡answer¿ and ¡/answer¿ tags to train the model for direct question answering. To explicitly teach the model the reasoning process, we apply targeted supervision for the penalty prediction task (q4) only. For this question, the training target includes the full human-authored CoT as supervised reasoning steps within `<think>...reasoning steps...</think>`, `<answer>...final answer...</answer>`. For the distinct task of free-form explanation (q6), we use a separate format where the target output is the full human rationale enclosed within `<Explanation>...explanations...</Explanation>`, signaling an explanation task. This approach enables a targeted, multi-task learning process, allowing the model to simultaneously learn direct answering, step-by-step reasoning, and detailed explanation generation.

## B.3   SUPERVISED FINE-TUNING (SFT)

Following the methodology of recent work in large-model alignment (Guo et al., 2025; Liu et al., 2025), we perform a "cold-start" SFT phase. This initial step serves two primary objectives in our experimental design.

First, it directly validates our benchmark's efficacy as a resource for supervised learning. By fine-tuning the model on 10% of the SportsImage data with 6 epochs, we demonstrate that our high-quality annotations can lead to performance gains over the zero-shot baselines, as shown in our results (Table 2).

Second, this SFT phase is used for preparing the model for the more intensive RL training. This aims to teach the model the format of our questions and the fundamental patterns present in the CoT rationales before the more intensive RL phase.

## B.4   REINFORCEMENT LEARNING

**Group Relative Policy Optimization:**   During the reinforcement learning phase, we employ the Group Relative Policy Optimization (GRPO) Shao et al. (2024b), which has been commonly used in reinforcement learning for model training across multiple domains Guo et al. (2025); Liu et al. (2025); Lai et al. (2025).

For each prompt $q$, we sample a group of $G$ responses $\{o_i\}_{i=1}^{G} \sim \pi_{\text{old}}(\cdot \mid q)$ and obtain rewards $\{r_i\}_{i=1}^{G}$. We define the advantage $A_i$ as follows:

$$A_i = \frac{r_i - \mu_r}{\sigma_r},$$

where $\mu_r, \sigma_r$ are the mean and standard deviation of $\{r_i\}$, and let $\rho_i = \dfrac{\pi_\theta(o_i \mid q)}{\pi_{\mathrm{old}}(o_i \mid q)}$. GRPO maximizes the following objective:

$$\mathcal{J}_{\mathrm{GRPO}}(\theta) = \mathbb{E}\left[\frac{1}{G}\sum_{i=1}^{G}\min\Big(\rho_i A_i,\ \mathrm{clip}(\rho_i,\ 1-\varepsilon,\ 1+\varepsilon)\,A_i\Big)\ -\ \beta\,D_{\mathrm{KL}}\big(\pi_\theta(\cdot \mid q)\,\|\,\pi_{\mathrm{ref}}(\cdot \mid q)\big)\right].$$

(1)

**Reward Function Design:** Our reward function $R(o|q)$ is a standard weighted combination of a correctness reward $R_{\mathrm{correct}}$ and a format incentive $R_{\mathrm{format}}$:

$$R(o|q) = 1.0 \cdot R_{\mathrm{correct}}(o|q) + 0.5 \cdot R_{\mathrm{format}}(o).$$

(2)

The correctness reward, $R_{\mathrm{correct}}$, is task-dependent. For the standard QA tasks (q1-q5), we use an LLM-as-Judge ( DeepSeek-V3.1 (Liu et al., 2024)) to provide a binary reward (The template can be found in Appendix B). If the output within the `<answer>...</answer>` is semantically consistent with the standard answer, a reward of 1 is assigned; otherwise, the reward is 0. For the explicit visual grounding task (q7), the reward is derived from the Generalized Intersection over Union (GIoU) (Rezatofighi et al., 2019). We opt for GIoU as the reward signal because, unlike standard IoU, it provides a meaningful gradient even for non-overlapping boxes, serving as a more effective training signal (Rezatofighi et al., 2019). Since the GIoU metric ranges from -1 to 1, we map it to a valid reward signal between 0 and 1 by applying the transformation: $(\mathrm{GIoU} + 1)/2$. The second component, $R_{\mathrm{format}}$, provides a format incentive for the reasoning task. A score of 1 is awarded if the output contains exactly one `<think>` block, one `</think>` block, one `<answer>` block, and one `</answer>`, and 0 otherwise. This reward is not applied to other tasks.

We did not include the Explanation questions (Q4) in the RL training phase, as their results are difficult to evaluate due to a lack of specific metrics for sports.

We conducted the GRPO-based reinforcement learning on 4xH20 GPUs.

## C  PROMPT

### C.1  INFERENCE PROMPT

You are an expert sports assistant with advanced multi-sport knowledge and image/video analysis capabilities for detecting tactics, fouls, and infractions.

Figure 4: System Prompt format for Answer Generation

You will receive a sport video. You need to answer the question. {\$Question}
You need to output your response based on the following instructions. Instruction: Valid output schemas (choose exactly ONE based on the question intent):
1) Explanation-only (when the question explicitly asks "why/explain"):
`<Explanation>...</Explanation>`
2) All other types of questions:
`<think>...</think><answer>...</answer>`

Figure 5: System Prompt format for Answer Generation

### C.2  TRAINING PROMPT

You are an expert sports assistant with advanced multi-sport knowledge and image/video analysis capabilities for detecting tactics, fouls, and infraction.

Figure 6: System Prompt format for Q1 - Q7 SFT training set

Answer the question after `<answer>` and end in `</answer>`. You don't need to output any reasoning process. Imaging you are an expert in sport, you will receive a picture of the sport scene. You need to answer the question. Is there a rule infraction (foul or infraction)? you can only answer "yes" or "no".
Output Example 1 for answer part:
`<answer> yes </answer>`
Output Example 2 for answer part:
`<answer> no </answer>`

Figure 7: User Prompt format for Q2 SFT training set

Answer the question after `<answer>` and end in `</answer>`. You don't need to output any reasoning process. Imaging you are an expert in sport, you will receive a picture of the sport scene. You need to answer the question. If a rule infraction (foul or infraction) occurred, what was the specific type? you only need to output the specific rule infraction (foul or infraction) type.
You need to output your answer after `<answer>` and end in `</answer>`.

Figure 8: User Prompt format for Q3 SFT training set

First, think between `<think>` and `</think>`, answer the question after `<answer>` and end in `</answer>`.
If a rule infraction (foul or violation) occurred, what is the resulting penalty?
You need to output your answer after `<answer>` and end in `</answer>`

Figure 9: User Prompt format for Q4 SFT training set

Answer the question after `<answer>` and end in `</answer>`. You don't need to output any reasoning process. Imaging you are an expert in sport, you will receive a picture of the sport scene. You need to answer the question.
If a rule infraction occurs in the image, find and locate it, and output the coordinates. If there is no rule infraction, locate the area that makes the most sense to determine whether a foul has been committed.
The coordinates must be written in the form: [x1, y1, x2, y2].
- x1 = left boundary (minimum x value, in pixels)
- y1 = top boundary (minimum y value, in pixels)
- x2 = right boundary (maximum x value, in pixels)
- y2 = bottom boundary (maximum y value, in pixels)
This bounding box uniquely defines a rectangle region.
From these four values, the four corners can be derived as:
(x1, y1) top-left, (x2, y1) top-right, (x2, y2) bottom-right, (x1, y2) bottom-left.
You can only output reasoning process between `<think>` and `</think>`. You need to output your answer after `<answer>` and end in `</answer>`.
Output Example for answer part: (do NOT copy the numbers, just follow the structure):
`<answer> [x1, y1, x2, y2] </answer>`

Figure 10: User Prompt format for Q5 SFT training set

Answer the question after `<answer>` and end in `</answer>`. You don't need to output any reasoning process.
What offensive formation or play is shown in this image? You need to output your answer after `<answer>` and end in `</answer>`.

Figure 11: User Prompt format for Q6 SFT training set

Answer the question after `<answer>` and end in `</answer>`. You don't need to output any reasoning process.
What defensive formation or play is shown in this image? You need to output your answer after `<answer>` and end in `</answer>`.

Figure 12: User Prompt format for Q7 SFT training set

{{ content — trim }}
You are an expert sports assistant with advanced multi-sport knowledge and image/video analysis capabilities for detecting tactics, fouls, and infractions.

Answer the question only between `<answer>` and `</answer>`. Do not output any internal reasoning, chain-of-thought, or content between `<think>` tags. If helpful, you may include a single-sentence clarification between `<explanation>` and `</explanation>` after the answer; otherwise omit it.

What defensive formation or play is shown in this image? Output your final answer only between `<answer>` and `</answer>`.

Figure 13: System Prompt format for RL training set

Please think about this question as if you were a human pondering deeply. Engage in an internal dialogue using expressions such as `"let me think"`, `"wait"`, `"Hmm"`, `"oh, I see"`, `"let's break it down"`, etc., or other natural language thought expressions. It's encouraged to include self-reflection or verification in the reasoning process. Provide your detailed reasoning between the `<think>` and `</think>` tags, and then give your final answer between the `<answer>` and `</answer>` tags.

Figure 14: User Prompt format for RL training set

Below are two answers to a question. [Question] is the question, [Standard Answer] is the standard answer, and [Model_answer] is the answer extracted from a model's output.

Determine whether these two answers are consistent in meaning. If they express essentially the same conclusion, treat them as consistent. Synonymous wording counts as consistent (e.g., "pink" and "it is pink"). Focus on the final English conclusion; ignore length, style, or reasoning details.

Output exactly one line:
Judgement: 1
or
Judgement: 0

Acceptance criteria (0/1):

Example:

""""""
[Question]: Is there a rule violation?
[Standard Answer]: Yes, there is a rule violation in the image.
[Model_answer]: Yes.
Judgement: 1
"""""",

""""""
[Question]: If a violation occurred, what was the specific type?
[Standard Answer]: Free Hand Touches the Playing Surface Violation.
[Model_answer]: Free hand touching the playing surface.
Judgement: 1
"""""",

""""""
[Question]: If a violation occurred, what is the resulting penalty?
[Standard Answer]: Point to the opponent.
[Model_answer]: The opponent is awarded the point.
Judgement: 1
"""""",

""""""
[Question]: Is there a rule violation?
[Standard Answer]: No, there is no rule violation in the image.
[Model_answer]: No.
Judgement: 1
"""""",

""""""
[Question]: If a violation occurred, what was the specific type?
[Standard Answer]: N/A
[Model_answer]: N/A
Judgement: 1
"""""",

"""
[Question]: If a violation occurred, what is the resulting penalty?
[Standard Answer]: No penalty.
[Model_answer]: No penalty.
Judgement: 1
""",

"""
[Question]: Is there a rule violation?
[Standard Answer]: Yes, there is a rule violation in the image.
[Model_answer]: No.
Judgement: 0
""",

"""
[Question]: If a violation occurred, what was the specific type?
[Standard Answer]: Net Touch.
[Model_answer]: Free Hand Touches the Playing Surface Violation.
Judgement: 0
""",

"""
[Question]: If a violation occurred, what is the resulting penalty?
[Standard Answer]: Point to the opponent.
[Model_answer]: Replay (let).
Judgement: 0
""",

"""
[Question]: Is there a rule violation?
[Standard Answer]: Yes, there is a rule violation in the image.
[Model_answer]: Yes.
Judgement: 1
""",

"""
[Question]: If a violation occurred, what was the specific type?
[Standard Answer]: Service Let.
[Model_answer]: Let serve.
Judgement: 1
""",

"""
[Question]: If a violation occurred, what is the resulting penalty?
[Standard Answer]: Let (replay).
[Model_answer]: Re-serve; the point is replayed.
Judgement: 1
""",

"""
[Question]: Is there a rule violation?
[Standard Answer]: Yes, there is a rule violation in the image.
[Model_answer]: Yes.
Judgement: 1
""",

"""
[Question]: If a violation occurred, what was the specific type?
[Standard Answer]: Service Fault.
[Model_answer]: Illegal serve (fault).
Judgement: 1
""",

"""
[Question]: If a violation occurred, what is the resulting penalty?
[Standard Answer]: Point to the opponent.
[Model_answer]: The receiver scores the point.
Judgement: 1
""",

"""
[Question]: Is there a rule violation?
[Standard Answer]: No, there is no rule violation in the image.
[Model_answer]: Yes.
Judgement: 0
""",

"""
[Question]: If a violation occurred, what was the specific type?
[Standard Answer]: N/A
[Model_answer]: Net Touch.
Judgement: 0
""",

"""
[Question]: If a violation occurred, what is the resulting penalty?
[Standard Answer]: No penalty.
[Model_answer]: Point to the opponent.
Judgement: 0
""",

Figure 15: Reward Prompt format for RL training set

## C.3 Evaluation Prompt Template

You are a scoring assistant for sports questions.

Rules

Compare the semantic meaning of "ground_truth" and "model_response" and assign a similarity score between 0 (completely different) and 1 (identical).

Output
Return ONLY one line:(replace the score to the real number)
```
<answer>score</answer>
```

Figure 16: System Prompt format for LLM-as-judge answer

Now is your turn
- sport: {$Sport}
- question: {$Question}
- ground_truth: {$Truth}
- model_response: {$ModelAnswer}

Figure 17: User Prompt format for LLM-as-judge answer

# D   ERROR ANALYSIS

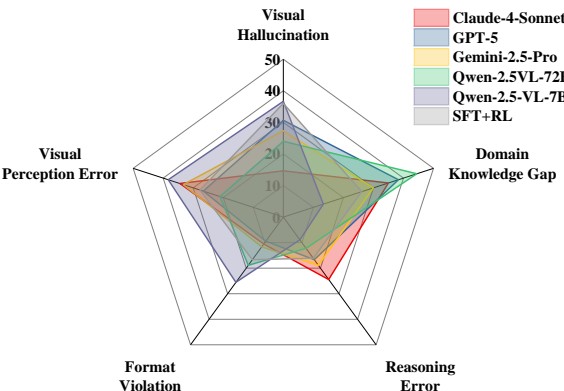

Figure 18: Error type distribution across different MLLMs on overall SportsImage tasks.

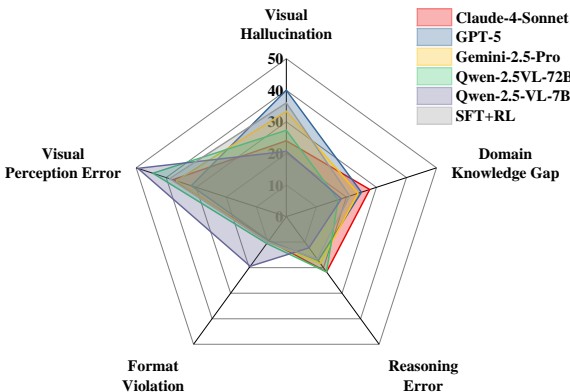

Figure 19: Error type distribution across different MLLMs on overall SportsVideo tasks.

To support the findings in Result Section, we provide the detailed statistical breakdown of error types separated by modality. Figure 18 and Figure 19 present the error distribution for SportsImage and SportsVideo, respectively.

## D.1   MODALITY-SPECIFIC OBSERVATIONS

**SportsVideo: The Perception Bottleneck.** As shown in Figure 19, video tasks are overwhelmingly dominated by **Visual Perception Errors** and **Visual Hallucinations**. For instance, the Qwen-2.5-VL-7B Base model exhibits a **49.33%** perception error rate in video, compared to 36.00% in images. Across all models, the combined rate of perception and hallucination errors frequently exceeds 60-70%. This indicates that the primary bottleneck in video is the inability to process fine-grained temporal dynamics. Models fail to correctly perceive the event, often preventing them from reaching the stage of rule application, resulting in an artificially low domain knowledge gap.

**SportsImage: The Knowledge Mapping Gap.** As shown in Figure 18, static images, where temporal ambiguity is removed, the error distribution shifts significantly. Stronger models show a sharp spike in **Domain Knowledge Gaps**. For example, Qwen-2.5VL-72B's knowledge gap rises from 10.67% in video to **43.33%** in images. Similarly, Claude-4-Sonnet's knowledge gap increases from 23.33% (Video) to **32.00%** (Image). This implies that when models can perceive the visual evidence more clearly, they still fail to map that evidence to the correct sports rule, validating the difficulty of the reasoning task itself.

## D.2   CASE STUDY

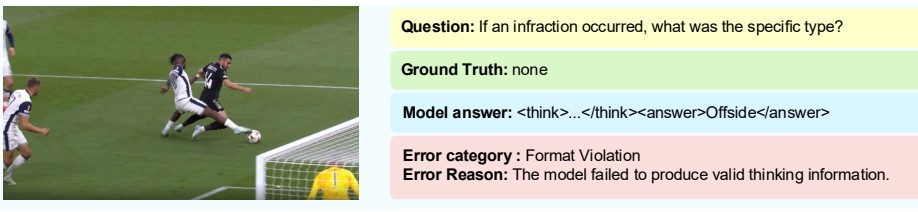

**Question:** If an infraction occurred, what was the specific type?

**Ground Truth:** none

**Model answer:** <think>...</think><answer>Offside</answer>

**Error category :** Format Violation
**Error Reason:** The model failed to produce valid thinking information.

Figure 20: Example of Format Violation

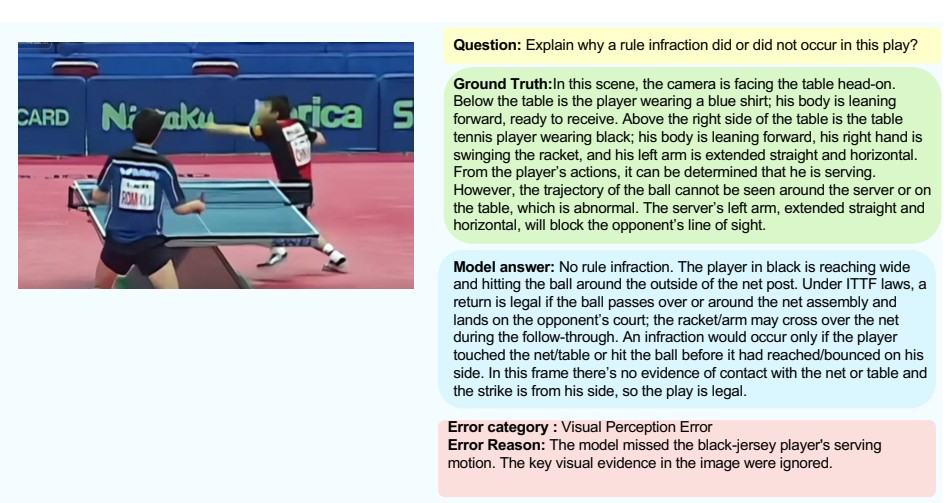

**Question:** Explain why a rule infraction did or did not occur in this play?

**Ground Truth:** In this scene, the camera is facing the table head-on. Below the table is the player wearing a blue shirt; his body is leaning forward, ready to receive. Above the right side of the table is the table tennis player wearing black; his body is leaning forward, his right hand is swinging the racket, and his left arm is extended straight and horizontal. From the player's actions, it can be determined that he is serving. However, the trajectory of the ball cannot be seen around the server or on the table, which is abnormal. The server's left arm, extended straight and horizontal, will block the opponent's line of sight.

**Model answer:** No rule infraction. The player in black is reaching wide and hitting the ball around the outside of the net post. Under ITTF laws, a return is legal if the ball passes over or around the net assembly and lands on the opponent's court; the racket/arm may cross over the net during the follow-through. An infraction would occur only if the player touched the net/table or hit the ball before it had reached/bounced on his side. In this frame there's no evidence of contact with the net or table and the strike is from his side, so the play is legal.

**Error category :** Visual Perception Error
**Error Reason:** The model missed the black-jersey player's serving motion. The key visual evidence in the image were ignored.

Figure 21: Example of Visual Perception Error

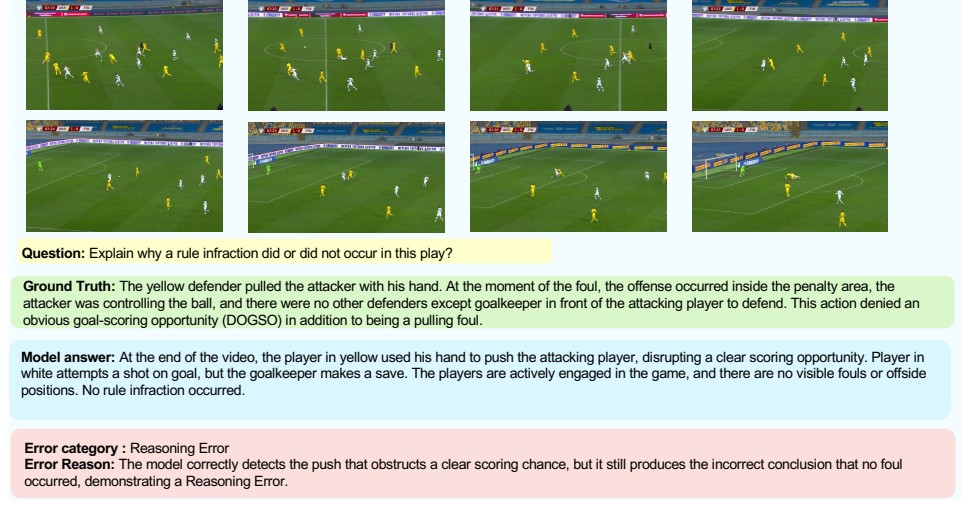

**Question:** Explain why a rule infraction did or did not occur in this play?

**Ground Truth:** The yellow defender pulled the attacker with his hand. At the moment of the foul, the offense occurred inside the penalty area, the attacker was controlling the ball, and there were no other defenders except goalkeeper in front of the attacking player to defend. This action denied an obvious goal-scoring opportunity (DOGSO) in addition to being a pulling foul.

**Model answer:** At the end of the video, the player in yellow used his hand to push the attacking player, disrupting a clear scoring opportunity. Player in white attempts a shot on goal, but the goalkeeper makes a save. The players are actively engaged in the game, and there are no visible fouls or offside positions. No rule infraction occurred.

**Error category :** Reasoning Error
**Error Reason:** The model correctly detects the push that obstructs a clear scoring chance, but it still produces the incorrect conclusion that no foul occurred, demonstrating a Reasoning Error.

Figure 22: Example of Reasoning Error

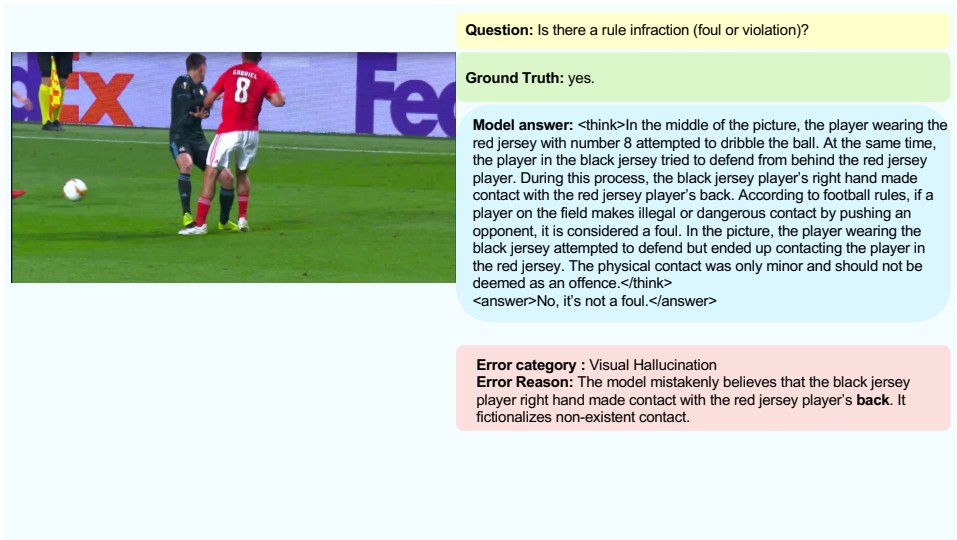

Figure 23: Example of Visual Hallucination

# E  CORRELATION MAP BETWEEN HUMAN AND LLMS

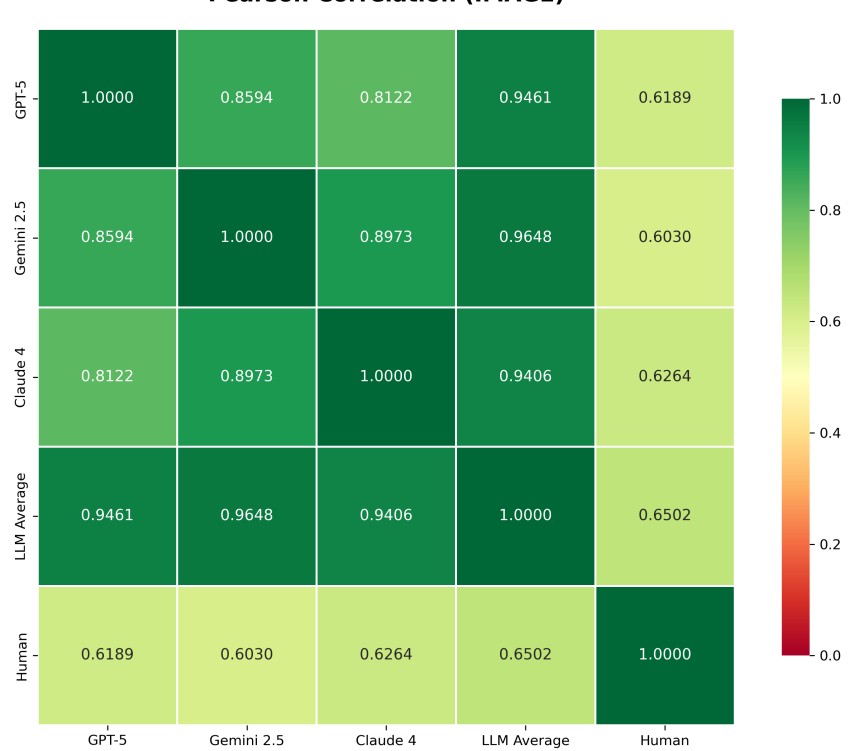

Figure 24: Pearson Correlation in image tasks.

To assess the reliability of our LLM-as-Judge framework, we conducted a **Human Verification Study** on a stratified subset of the test set. This study involved 660 randomized samples (360 from SportsImage and 300 from SportsVideo), covering explanation tasks. Expert human annotators

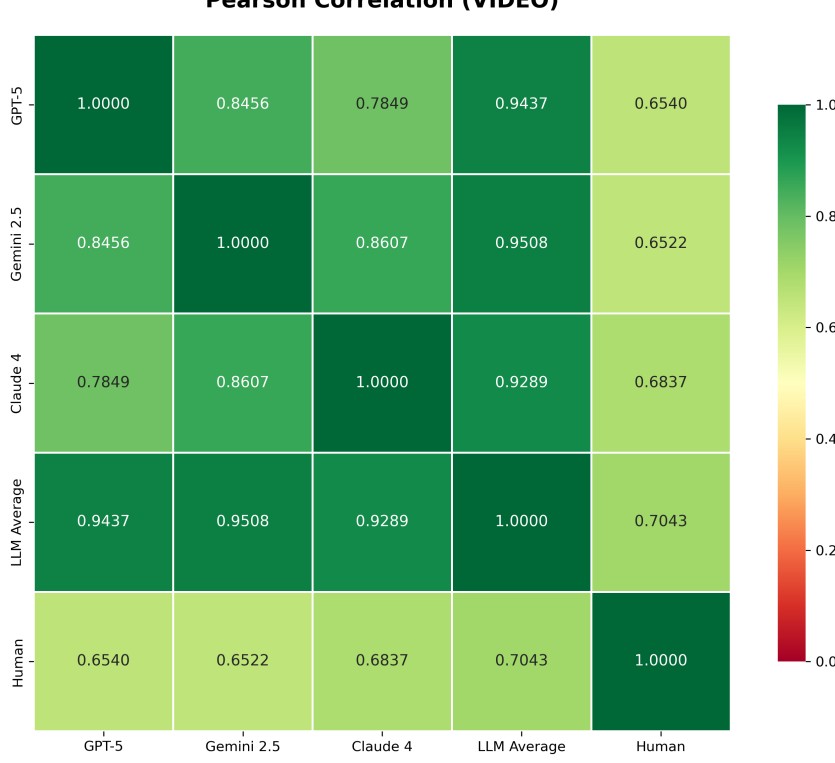

Figure 25: Pearson Correlation in video tasks.

scored the model responses following the exact same instructions provided to the LLM judges, operating blindly without knowledge of the model sources.

### E.1 CORRELATION ANALYSIS

We calculated the Pearson correlation coefficient between the human scores and the scores assigned by the three LLM judges (GPT-5, Gemini 2.5 Pro, Claude 4 Sonnet) as well as their ensemble average. The results are visualized in Figure 24 and Figure 25

**Internal Consistency.** First, we observe high internal consistency among the three proprietary models (Pearson correlation ranging from 0.81 to 0.96). This indicates that frontier MLLMs share a robust, stable internal standard for evaluating sports reasoning, reducing concerns about randomness or model-specific hallucinations in grading.

**Alignment with Human Judgment.** Crucially, the **LLM Average Score** demonstrates a stronger correlation with human experts ($r \approx 0.65$ for Image, $r \approx 0.70$ for Video) than any single LLM judge. For instance, while individual models occasionally diverge from human ratings due to specific biases, the ensemble approach effectively mitigates this variance. This confirms that averaging scores from multiple top-tier models serves as a reliable proxy for human evaluation in this benchmark.

### E.2 SCOPE OF EVALUATION METRIC

It is important to clarify the objective of this evaluation setup. **Our primary goal is to validate the relative difficulty of the SportR benchmark and the performance hierarchy of current models, rather than to propose a novel, perfect sports evaluation metric.** The analysis confirms that our metric is sufficiently robust to distinguish between model capabilities and to establish a reliable comparison, which fulfills the purpose of this benchmarking study.

# F EXAMPLES

## F.1 SPORTSIMAGE EXAMPLES

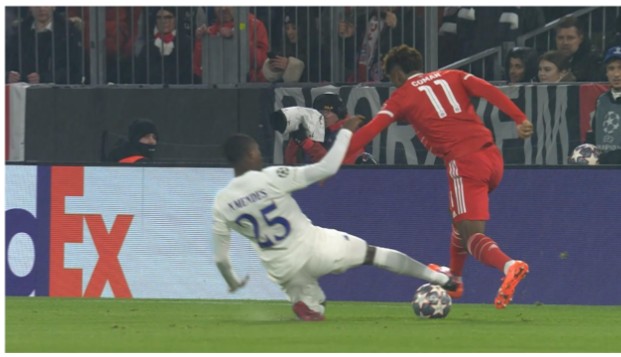

Question: Is there a rule infraction in the image?
Answer: Yes, there is a rule violation in the image

Figure 26: Example of Q1

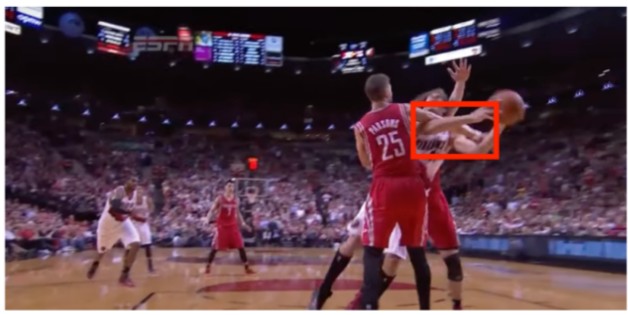

Question:
If a rule infraction occurs in the image, find and locate it, and output the coordinates.

Answer:
[2072, 703, 2271, 833]

Figure 27: Example of Q5

## F.2 SPORTSVIDEO EXAMPLES



Question:
Explain why a rule infraction did or did not occur in this play?

Answer: The purple jersey player number 93 is rushing the white jersey player number 16. The white jersey number 16 is throwing the ball to the left field where shows no obvious eligible receiver. This forward pass choice intentionally initiated by the passer 16 is considered without a realistic chance of completion because of pressure from the defense.

Figure 28: Example of Q10

## F.3 SPORTSVIDEO EXAMPLES

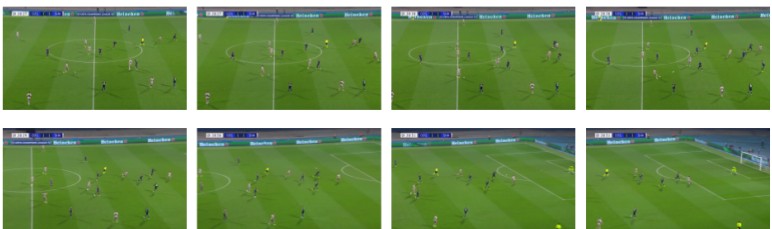

Question: If a infraction occurred in the video, what is the resulting penalty?
Answer:
In the video, the grey team is attacking while the blue team is defending. During the attack, the player in the grey which is ahead of the three blue players is in an offside position, as he is nearer to the opponents' goal line than both the ball and the player in blue jersey who is considered as the second-last opponent. According to football rules, a player in an offside position when the ball is played by a teammate is only penalized if they become actively involved in the game by interfering with play, interfering with an opponent, or gaining an advantage from being in that position, with an exception being made if the player receives the ball from an opponent who has made a deliberate play. In the video, this grey jersey player is in an offside position when his teammate pass the ball. He clearly attempts to play the ball in the offside position. Therefore, it is considered an offside offence. Since it's an offside foul, an indirect free kick will be awarded.

Figure 29: Example of Q11

