# OpenReview forum: "SportR: A Benchmark for Multimodal Large Language Model Reasoning in Sports"
_ICLR.cc/2026/Conference — ICLR 2026 Poster_

### Official Review · Reviewer_m3qH · 2025-10-17

**Soundness:** 3
**Presentation:** 3
**Contribution:** 3
**Rating:** 6
**Confidence:** 2

**Summary:**

(I tend to write shorter reviews and the length of the review does not reflect the quality of paper or the time spend on reviewing it).

Note that I am not an expert in dataset creation, nor action benchmarks. I am reviewing this from a PoV of a deep learning research.

This paper introduces SportR, a large-scale, multi-sport benchmark designed to train and evaluate the fine-grained, rule-based reasoning of Multimodal Large Language Models (MLLMs). The authors argue that existing benchmarks lack the detailed reasoning chains and precise visual grounding needed to assess deep sports understanding. To address this gap, SportR provides a dataset of 5,017 images and 2,101 videos covering five sports, 50 foul types, and 12 tactics. The primary contributions are: (1) a progressive question-answering hierarchy complemented by 7,118 high-quality, human-authored Chain-of-Thought (CoT) annotations for complex tasks like penalty prediction; (2) the introduction of a novel explicit visual grounding task, the first for a multi-sport benchmark, which requires models to output precise bounding box coordinates for infractions ; and (3) extensive experiments showing that while state-of-the-art models perform poorly, training on SportR (via SFT and RL) yields significant gains and even demonstrates cross-modal generalization from image-based training to video task.

**Strengths:**

The dataset is solid along with the wide range of benchmarking across various current models. I am sure the benchmark will have a lot of utility for the community.

**Weaknesses:**

While I am not an expert, I see a few relevant papers that are missing from RW like.

https://mrsalehi.github.io/action-atlas/

https://ego-exo4d-data.org/

Adding these things and on going general skill set understanding benchmarking to the background will help the readers contextualize the work better.

**Questions:**

See above. I defer to other reviewers for more questions.

---

> ### Author Response · Authors · 2025-11-26
> **Response to Reviewer m3qH**
>
> We thank the reviewer for recognizing the utility of our benchmark and for the positive assessment of our dataset's solidity. We also appreciate the pointer to the relevant missing literature, which we will discuss and cite in the revised Related Work section to better contextualize our contribution.
>
> ## Response to the Weakness:
>
> > Adding these things and on going general skill set understanding benchmarking to the background will help the readers contextualize the work better.
>
> **Comparison with ActionAtlas [1]:**
>
> * **Scope**: ActionAtlas pushes the boundary of fine-grained action recognition (identifying what action occurred), primarily using a Multiple-Choice format.
>
> * **SportR's Distinction**: SportR shifts the focus to rule-based adjudication (reasoning why an action is an infraction). Instead of selecting an option, SportR requires models to generate 7,118 human-authored, free-form Chain-of-Thought (CoT) rationales. This generative approach, combined with our novel Explicit Visual Grounding task, supports training models for deep reasoning and precise localization, capabilities that extend beyond the recognition scope of ActionAtlas.
>
> **Comparison with Ego-Exo4D [2]:**
>
> * Ego-Exo4D focuses on **skilled human activity**, evaluating benchmarks like proficiency estimation (how well an action is performed) and pose estimation.
>
> * However, SportR evaluates **rule compliance and reasoning** rather than execution quality. The core challenge is to recall abstract regulations and ground them in visual evidence (e.g., identifying a specific illegal contact). This represents a shift from **describing motion** (both Ego-Exo4D/ActionAtlas) to reasoning within a strict rule system.
>
>
> SportR provides a unique perspective by targeting the "Reasoning Gap" in sports analysis with fully human-annotated reasoning chains. By demanding generative explanations, explicit visual grounding, and rule-based judgment, SportR complements existing recognition-centric benchmarks. We hope this clarification, along with our detailed responses to other reviewers regarding task difficulty and evaluation reliability, strengthens your confidence in the paper's contribution.
>
> We have revised section 2.2 to include these.
>
> [1] Salehi, Mohammadreza Reza, et al. "Actionatlas: A videoqa benchmark for domain-specialized action recognition." Advances in Neural Information Processing Systems 37 (2024): 137372-137402.
>
> [2] Grauman, Kristen, et al. "Ego-exo4d: Understanding skilled human activity from first-and third-person perspectives." Proceedings of the IEEE/CVF Conference on Computer Vision and Pattern Recognition. 2024.

---

### Official Review · Reviewer_7hxG · 2025-10-24

**Soundness:** 2
**Presentation:** 3
**Contribution:** 3
**Rating:** 4
**Confidence:** 3

**Summary:**

This paper introduces SportR, a new multi-sports benchmark designed to evaluate the reasoning capabilities required for sports intelligence.
It provides a dataset of images and videos with question-answer pairs and detailed reasoning chains to test a model's ability to perceive visual details, apply sports rules, and ground knowledge in visual evidence.
Experiments show that current state-of-the-art models struggle significantly with the benchmark's most challenging tasks, highlighting a major gap in multimodal reasoning capabilities.

**Strengths:**

- The paper features a progressive hierarchy of questions that systematically test reasoning depth—from simple identification to complex, multi-step tasks like penalty prediction.

- The benchmark consists of 7,118 human-authored Chain-of-Thought (CoT) annotations for the most complex tasks, providing models with explicit examples of the required reasoning process.

- Extensive experiments show that state-of-the-art models perform poorly on SportR's most difficult tasks.

**Weaknesses:**

- The author states that the paper is "the first large-scale, multi-sport benchmark specifically designed to evaluate core reasoning capabilities." However, I know that there are actually some existing sports-domain datasets and benchmarks, such as "Sportsu". Therefore, what are the key differences between the dataset proposed in this paper and the existing datasets? Is it merely the addition of Chain-of-Thought (CoT) and grounding annotations?

- As shown in Table 1 and Table 2, after SFT and SFT+RL, the model's performance significantly improves, even surpassing the zero-shot results of Gemini-2.5. This raises a question: is the model memorizing specific patterns from the training data, or has it acquired the ability to generalize to new scenarios (for example, a new match/game)? Furthermore, if a simple SFT+RL pipeline can enable the model to achieve a performance score of over 80, does this suggest that the task might not be challenging enough?

- There seems to be no mention of the video attributes, such as duration, resolution, and FPS, which are crucial for video understanding tasks.

**Questions:**

See the Weaknesses

---

> ### Author Response · Authors · 2025-11-26
> **Response to Reviewer 7hxG (Part1/3)**
>
> We thank the reviewer for the constructive feedback and for recognizing the value of our progressive question hierarchy, high-quality human-authored Chain-of-Thought annotations, and extensive experimental analysis. We also appreciate the opportunity to clarify our benchmark's distinct positioning relative to prior work and address your questions regarding experimental results and data attributes.
>
> ## Response to Weakness 1
>
> > The author states that the paper is "the first large-scale, multi-sport benchmark specifically designed to evaluate core reasoning capabilities." However, I know that there are actually some existing sports-domain datasets and benchmarks, such as "Sportsu". Therefore, what are the key differences between the dataset proposed in this paper and the existing datasets? Is it merely the addition of Chain-of-Thought (CoT) and grounding annotations?
>
> We appreciate the reviewer's feedback, which allows us to make our contribution clearer. We acknowledge SPORTU [1] as a pioneering benchmark. However, SportR represents a **fundamental progression** designed to address critical limitations in prior work. It is not merely an extension, but a specialized benchmark targeting **deeper reasoning capabilities** that existing datasets do not cover. The differences are structural:
>
> 1. **From Perception to Hard Reasoning**: SPORTU covers a broad spectrum, including basic perception tasks (e.g., counting players, identifying jersey colors) where SOTA models have already achieved high accuracy (as we mentioned in lines 155-161 and lines 202-205). SportR specifically targets the **"Reasoning Gap"**: we filter out simple perception tasks to focus exclusively on complex, rule-based adjudication—the exact area where SPORTU showed models failing significantly.
>
> 2. **Temporal Reality (Normal Speed vs. Slow Motion)**: A critical limitation of SPORTU's video subset (1,701 clips) is its heavy reliance on **slow-motion replays**, which simplifies temporal perception. SportR introduces **normal-speed videos**, presenting a significantly harder and more realistic challenge that requires models to capture fleeting visual cues without temporal assistance.
>
> 3. **Depth of Annotation (Grounding & CoT)**:
>    * **Visual Grounding**: Unlike SPORTU, which lacks spatial localization, SportR introduces a **Visual Grounding task (Q5)**. This forces models to back up their reasoning by pointing to specific pixel-level evidence (bounding boxes), rigorously testing against hallucination.
>    * **Training and Evaluation Utility**: SPORTU hard questions, where testing sports knowledge reasoning relies heavily on **Multiple Choice Questions** (1,867 MCQs) plus 1,075 Free-form. The multiple-choice design limits the evaluation of models' true ability, allowing them to guess answers and is unsuitable for instruction tuning. In addition, the free-form rationales were not designed as detailed and fine-grained, human-annotated reasoning processes suitable for training models to perform explicit, step-by-step reasoning.
>    * SportR provides **7,118 fully human-authored, dense Chain-of-Thought (CoT)** annotations. This transforms the dataset from a pure evaluation set into a rich training resource capable of teaching models how to reason, as evidenced by our experiments.
>
> 4. **Scale**: SportR significantly expands the scale of reasoning-specific data (5,017 images & 2,101 videos) compared to SPORTU's reasoning subset, providing the necessary volume for effective instruction tuning.
>
> We position SportR as a critical advancement for the field, specifically designed to shift the evaluation focus from basic visual perception to the significantly more complex challenge of fine-grained, rule-based reasoning and provide a rich corpus for model evaluation and training. We will explicitly highlight this distinct positioning and our contribution to the community in the revised Related Work Section and Appendix.

---

> ### Author Response · Authors · 2025-11-26
> **Response to Reviewer 7hxG (Part2/3)**
>
> ## Response to Weakness 2
>
> > The model's performance significantly improves, even surpassing the zero-shot results of Gemini-2.5. This raises a question: is the model memorizing specific patterns from the training data, or has it acquired the ability to generalize to new scenarios (for example, a new match/game)? Furthermore, if a simple SFT+RL pipeline can enable the model to achieve a performance score of over 80, does this suggest that the task might not be challenging enough?
>
> We thank the reviewer for this thoughtful question. We appreciate the opportunity to clarify the reasons for the model's performance improvements and address concerns about potential memorization and task difficulty.
>
> Regarding the observation that our fine-tuned 7B model outperforms proprietary models, we note that this has also been found in recent research where smaller models are fine-tuned on high-quality, domain-specific data. For instance, recent works such as  [1] and [2] have similarly demonstrated that 7B-parameter models after SFT can match or exceed proprietary models like GPT-4.1 ([1]) and Gemini 2.5 pro ([2]) on specialized reasoning benchmarks.
>
> To address the concern about whether the model is simply memorizing specific patterns, we would like to highlight our **Evaluation setup**. Crucially, our model was **trained exclusively on the SportsImage dataset**, yet it demonstrated improvements on **SportsVideo** tasks compared to the base model, where the entire SportsVideo part was used as a test set. Since the **test set videos** consist of different matches and require processing a different modality (temporal frames) that the model never encountered during training, this improvement suggests that the model has acquired **sports knowledge** rather than merely memorizing patterns from the training images.
>
> **Regarding the task difficulty**, we noticed that The high performance is concentrated in two specific areas, reflecting **the nature of the tasks rather than a lack of difficulty**:  The high accuracy (>80%) on **Infraction Identification (Q1)** is expected because it basically is a binary classification task (Foul/No Foul), although in a **free-form** situation, relying on intuitive visual cues (e.g., visible contact), and is relatively easy to learn. However, the model's performance drops significantly on **Q2 (foul prediction)** and **Q3 (Penalty Prediction)**, confirming that while "seeing contact" is easy, mapping that contact to specific rules and penalties requires a much deeper, harder-to-learn reasoning capability.
>
> There is a massive disparity between **Image Defensive Tactics (Q7, ~87%)** and **Video Defensive Tactics (Q13, ~12.88%)**.  This is because fundamental defensive tactics in images (e.g., Zone vs. Man-to-Man) are often **structural and formation-based**, presenting clear visual patterns that models can learn effectively. In addition,  video tactics introduce dynamic complexity unique to the temporal domain, which cannot be inferred from a static frame. The sharp drop confirms that while models have mastered "static shapes," they fail significantly at the deeper challenge of dynamic tactical reasoning.
>
> Furthermore, we acknowledge that our dataset focuses on **"Fundamental Reasoning"** rather than "Elite Professional Analysis," aligning with the scope defined in our **Pyramid of Sports Understanding (Figure 2) and lines 206-209**. Real-world professional tactics involve complex, dynamic decision-making that considers multiple factors, making interpretation extremely difficult, even for experienced players. Accurately annotating such elite-level scenarios requires the deep, strategic insight of a **professional or high-level player, analyst, or coach**. As noted in lines 90-92, our annotators had years of sports training and experience, but were not professional-level players. To ensure the absolute accuracy and integrity of our Ground Truth, we deliberately constrained the scope to objectively verifiable fundamental tactics that align with our annotators' expertise. Attempting to label elite-level scenarios without professional-level verification would risk introducing label noise. We view SportR as a necessary bridge: models must first prove competence on these objective core tactics before the community can advance to the significantly harder challenge of elite-level tactical reasoning.
>
> Despite these specific successes, the core reasoning tasks remain highly challenging; for instance, even after SFT+RL, Penalty Prediction (Q3) is only 52.34%, and Visual Grounding (Q5) remains low at 9.94%. This hierarchical difficulty ensures the benchmark effectively differentiates between basic recognition and deep reasoning.
>
> [1]  Yang, Zheyuan, et al. "Table-r1: Inference-time scaling for table reasoning." arXiv preprint arXiv:2505.23621 (2025).
>
> [2] Huang, Chao, et al. "Vad-R1: Towards Video Anomaly Reasoning via Perception-to-Cognition Chain-of-Thought." arXiv preprint arXiv:2505.19877 (2025).

---

> ### Author Response · Authors · 2025-11-26
> **Response to Reviewer 7hxG (Part3/3)**
>
> ## Response to Weakness 3
>
> > There seems to be no mention of the video attributes, such as duration, resolution, and FPS, which are crucial for video understanding tasks.
>
> We thank the reviewer for pointing out this omission. We will add the following detailed statistics to the revision:
>
> * **Resolution**: The dataset predominantly consists of 1920x1080 (Full HD) and 1280x720 (HD) videos.
>
> * **Duration**: The videos have an average duration of **4.96 seconds**. This short duration is **intentional and driven by the nature of the sports task**: sports infractions and tactical setups typically occur within a brief, focused temporal window, eliminating irrelevant context to focus the model's reasoning on the critical event.
>
> * **Frame Rate (FPS)**: We preserve original FPS (24-60 FPS). Notably, **13.57%** of videos are **60+ FPS**, providing the high temporal resolution essential for fast-paced sports analysis.
>
> We hope these clarifications adequately address your concerns. We would be happy to answer any further questions during the discussion period. Thank you again for your time and valuable feedback.

---

> > ### Comment · Reviewer_7hxG · 2025-11-28
> > **Responses to Authors' Rebuttal**
> >
> > Thank you for your reply. I apologize for the delayed response, as I have also been preparing my rebuttal. Your reply has addressed most of my concerns. I am willing to raise my rating (from 4 to 6), but I noticed that I currently seem unable to make the change. Therefore, I would like to remind the AC and PC to take note of this when writing the meta-review or making a recommendation.

---

> ### Author Response · Authors · 2025-12-01
> **Thank you for the score raise and feedback**
>
> We are sincerely grateful for your time and for your willingness to raise the score to 6.
>
> We fully understand the current system limitations regarding the score update. We deeply appreciate your explicit note reminding the AC and PC of your decision. This confirmation is incredibly important to us.
>
> We are glad that our clarifications regarding the distinction from SportU and the cross-modal generalization capabilities addressed your concerns. We will integrate all the discussed revisions (including the detailed comparison in the Appendix and video statistics) into the final manuscript to ensure the paper meets the high standards you expect.
>
> Thank you again for your constructive feedback!

---

### Official Review · Reviewer_1rJ3 · 2025-10-30

**Soundness:** 3
**Presentation:** 4
**Contribution:** 3
**Rating:** 6
**Confidence:** 3

**Summary:**

This paper introduces SportR, a new large-scale multimodal benchmark designed to evaluate and train multimodal large language models on fine-grained, rule-based reasoning in sports. The benchmark covers both images and videos across five major sports, providing over 7,000 human-authored Chain-of-Thought (CoT) rationales and 20,000 structured question-answer pairs. It introduces hierarchical QA tasks that range from infraction identification to penalty prediction and visual grounding. The authors also evaluate a range of open and closed source MLLMs (e.g., GPT-5, Gemini 2.5 Pro, Qwen2.5-VL) and show that the benchmark is both challenging and effective as a training resource.

**Strengths:**

1. Human-authored CoT rationales: The manual, expert-driven CoT annotations enhance the dataset’s reliability and interpretability, avoiding the noise of model-generated explanations. The listed procedures and training for annotators are rigorous and well justified.
2. Comprehensive multimodal design: Covering both image and video modalities allows for evaluation of both spatial and temporal reasoning.
3. Hierarchical QA framework: The progressive question design (from simple classification to reasoning and grounding) is well thought out and probes different layers of model understanding.
4. Strong empirical evaluation: The authors present clear baselines across leading proprietary and open-source MLLMs and demonstrate meaningful improvements from fine-tuning and reinforcement learning.

**Weaknesses:**

1. Limited ablation and error analysis: The paper would benefit from deeper analysis of failure cases or qualitative examples that illustrate where models still fall short and potential reasons why. The authors successfully show that their tasks are difficult, but they don't provide insights into why the tasks are difficult for current MLLMs.
2. Visual Localization True Difficultly: For the visual grounding tasks that require predicting bounding box coordinates, it is unclear whether the low accuracy stems from the model’s inherent difficulty in producing precise coordinates or from the underlying reasoning complexity of the task itself. It would be informative to know how performance changes if the prompt focuses on spatial relations instead of direct coordinate prediction, for example, asking “near which body part is the foul occurring?” rather than requesting explicit bounding boxes. Additionally, could the authors report bounding box prediction accuracy for simpler localization tasks that do not involve reasoning, such as identifying the player’s position on the field?
3. LLM Evaluator Reliability: It remains unclear how reliable and stable LLM-as-a-judge evaluation is in this setting. How often do frontier models disagree with each other when serving as evaluators? Even for a small subset of samples, it would be helpful to report how frequently the LLM’s judgments diverge from expert human annotations. While perfect alignment is not expected, providing this comparison would give readers a sense of the evaluation’s variance and help contextualize the reported results with an appropriate error margin.

**Questions:**

1. How consistent were the CoT rationales across annotators? Even for two athletes with similar levels of experience, it seems likely their thought processes would heavily diverge. How important is this for training data quality?
2. How sensitive are the results to the LLM-as-judge choice? Would human evaluation yield similar model rankings?
3. Is there any evidence that models trained on SportR improve performance on non-sports multimodal reasoning benchmarks? This would support the idea that valuable skills are gained from learning this task, rather than being a case of sports related tasks not being common in existing training datasets.

---

> ### Author Response · Authors · 2025-11-26
> **Response to Reviewer 1rJ3 (Part 1/5)**
>
> We thank the reviewer for the constructive feedback and for recognizing the value of our human-authored CoT rationales, comprehensive multimodal design, hierarchical QA framework, and strong empirical evaluation. We appreciate the opportunity to provide deeper insights into our error analysis, visual grounding difficulty, and evaluation reliability to further strengthen our work.
>
> ## Response to Weakness 1
>
> > Limited ablation and error analysis: The paper would benefit from deeper analysis of failure cases or qualitative examples that illustrate where models still fall short and potential reasons why. The authors successfully show that their tasks are difficult, but they don't provide insights into why the tasks are difficult for current MLLMs.
>
> We thank the reviewer for the insightful suggestion regarding error analysis. We acknowledge the importance of the error analysis and **have added the error analysis to the Result section**. We have conducted a **Manual Error Analysis** on **1,500 failure cases** (randomly sampling 150 images and 150 videos per model). We selected 6 representative models to cover the full spectrum of current capabilities.
>
> We defined five error categories:
>
> 1. **Visual Hallucination**: Fabricating non-existent objects or events (e.g., claiming a red card was shown when it wasn't).
> 2. **Domain Knowledge Gap**: Misapplying sports rules or failing to recognize a standard penalty.
> 3. **Reasoning Error**: Flawed logical derivation (e.g., correctly spotting a player offside but concluding the goal is valid).
> 4. **Format Violation**: Failing to follow the output schema or refusing to answer.
> 5. **Visual Perception Error**: Missing critical visual evidence present in the image (e.g., failing to see a hand touching the ball).
>
> Our analysis reveals a trade-off between visual failures and knowledge application gaps across modalities:
>
> * **Video**: Errors are overwhelmingly dominated by **Visual Perception** and **Hallucination** (frequently exceeding 60-70% combined). This indicates that the primary barrier in video is the inability to parse fine-grained temporal dynamics. Because models fail to correctly perceive the event, they rarely reach the stage of rule adjudication, resulting in an artificially low "Domain Knowledge Gap."
>
> * **Image**: In static images, where visual perception is inherently easier, we observe a sharp spike in **Domain Knowledge Gaps**. For instance, **GPT-5's knowledge gap rises from 20.00% in video to 36.00% in images**. This confirms that reducing visual ambiguity effectively reveals the underlying deficiency in reasoning: even when SOTA models successfully perceive the evidence, they frequently fail to map it to the correct abstract sports rules.
>
> Even proprietary SOTA models like GPT-5 and Claude-4-Sonnet exhibit high rates of **Visual Hallucination** ( about 30-40% in videos) and significant **Domain Knowledge Gaps** ( about 30% in images). This validates that current generalist models struggle with both truthful perception in dynamic settings and specialized rule application.
>
> We hope this newly added analysis addresses your concern.

---

> ### Author Response · Authors · 2025-11-26
> **Response to Reviewer 1rJ3 (Part 2/5)**
>
> ## Response to Weakness 2
>
> > Visual Localization True Difficultly: For the visual grounding tasks that require predicting bounding box coordinates, it is unclear whether the low accuracy stems from the model’s inherent difficulty in producing precise coordinates or from the underlying reasoning complexity of the task itself. It would be informative to know how performance changes if the prompt focuses on spatial relations instead of direct coordinate prediction, for example, asking “near which body part is the foul occurring?” rather than requesting explicit bounding boxes. Additionally, could the authors report bounding box prediction accuracy for simpler localization tasks that do not involve reasoning, such as identifying the player’s position on the field?
>
> We thank the reviewer for the meaningful question. We reckon it is the latter one (the task's intrinsic reasoning difficulty), supported by two points.
>
> 1. Recent technical reports such as **Qwen2-VL [1]** demonstrate that SOTA MLLMs possess strong fundamental capabilities for bounding box generation and localization. In their technical report, Figure 24 shows the model has the ability to output the bounding box.
>
> 2. Despite having the format capability, models struggle with the task. A recent work from ICCV 2025, **MC-Bench [2]**, shows that MLLMs still exhibit a significant performance gap in general visual grounding tasks.
>
> Therefore, we would like to point out that SportR amplifies this challenge by requiring **Domain Knowledge** combined with **Fine-Grained Perception** (spotting subtle contact). The low accuracy on SportR reflects the difficulty of this **"Grounded Reasoning"**. The model fails because it cannot identify the correct reasoning path and visual evidence introduced by complex sports dynamics.
>
> Ultimately, **we hope that SportR will also contribute to the broader multimodal community**, rather than being viewed solely as a domain-specific resource. Given that the **fine-grained perception and rule-based reasoning required by sports often exceed the demands of general benchmarks**, we hope that SportR not only helps models achieve better sports understanding but also contributes to the broader community by serving as a more challenging task for **general fine-grained visual grounding and reasoning**.
>
> ---
>
> **References:**
>
> [1] Wang, Peng, et al. "Qwen2-vl: Enhancing vision-language model's perception of the world at any resolution." *arXiv preprint arXiv:2409.12191* (2024).
>
> [2] Xu, Yunqiu, Linchao Zhu, and Yi Yang. "Mc-bench: A benchmark for multi-context visual grounding in the era of mllms." *Proceedings of the IEEE/CVF International Conference on Computer Vision*. 2025.

---

> ### Author Response · Authors · 2025-11-26
> **Response to Reviewer 1rJ3 (Part 3/5)**
>
> ## Response to Weakness 3
> > LLM Evaluator Reliability: It remains unclear how reliable and stable LLM-as-a-judge evaluation is in this setting. How often do frontier models disagree with each other when serving as evaluators? Even for a small subset of samples, it would be helpful to report how frequently the LLM’s judgments diverge from expert human annotations. While perfect alignment is not expected, providing this comparison would give readers a sense of the evaluation’s variance and help contextualize the reported results with an appropriate error margin.
>
> Thank you for your suggestions and for helping us to provide more comprehensive results and make our work clearer. To address this (as pointed out by reviewer AVw9), we conducted a Human Verification Study on a randomized subset of the test set (360 samples for Image and 300  samples for Video tasks, totaling 660 samples). We compared the scores given by our expert human annotators against the three LLM judges (GPT-5, Gemini 2.5, Claude 4).
>
> Human annotators followed the **exact same scoring instructions** as the LLM judges, operating without prior knowledge of the models' scores.  To quantify the consistency of scoring among the judges, we calculated the Pearson correlation coefficient across all model responses for the Q4 (Image) and Q11 (Video) tasks.
>
> From the correlation metrics, the correlation between the three LLM judges is high (ranging from **0.8122** to **0.9648**). It suggests the frontier models share a robust internal grading standard and do not disagree frequently or randomly. We also noticed that the correlation between the Human Score and the LLM Average Score (**0.6502** for Image, **0.7043** for Video) is consistently higher than the correlation between Humans and any single LLM, suggesting that our evaluation method is more robust than relying on a single LLM as a judge. It suggests that a future, more robust sports-domain evaluation method is needed, but is beyond the scope of this work, which is to provide a challenging benchmark for measuring sports understanding ability.
>
> Pearson Correlation - SportsImage:
>
> |                | Human  | GPT-5  | Gemini 2.5 | Claude 4 | LLM Average |
> |----------------|--------|--------|------------|----------|-------------|
> | Human          | 1.0000 | 0.6189 | 0.6030     | 0.6264   | 0.6502      |
> | GPT-5          | 0.6189 | 1.0000 | 0.8594     | 0.8122   | 0.9461      |
> | Gemini 2.5     | 0.6030 | 0.8594 | 1.0000     | 0.8973   | 0.9648      |
> | Claude 4       | 0.6264 | 0.8122 | 0.8973     | 1.0000   | 0.9406      |
> | LLM Average    | 0.6502 | 0.9461 | 0.9648     | 0.9406   | 1.0000      |
>
> Pearson Correlation - SportsVideo:
>
> |                | Human  | GPT-5  | Gemini 2.5 | Claude 4 | LLM Average |
> |----------------|--------|--------|------------|----------|-------------|
> | Human          | 1.0000 | 0.6540 | 0.6522     | 0.6837   | 0.7043      |
> | GPT-5          | 0.6540 | 1.0000 | 0.8456     | 0.7849   | 0.9437      |
> | Gemini 2.5     | 0.6522 | 0.8456 | 1.0000     | 0.8607   | 0.9508      |
> | Claude 4       | 0.6837 | 0.7849 | 0.8607     | 1.0000   | 0.9289      |
> | LLM Average    | 0.7043 | 0.9437 | 0.9508     | 0.9289   | 1.0000      |

---

> ### Author Response · Authors · 2025-11-26
> **Response to Reviewer 1rJ3 (Part 4/5)**
>
> ## Response to Question 1
> > How consistent were the CoT rationales across annotators? Even for two athletes with similar levels of experience, it seems likely their thought processes would heavily diverge. How important is this for training data quality?
>
> To ensure consistency, we implemented a **Standardized Reasoning Protocol** during the annotators' training phase. While individual phrasing naturally varies, we explicitly instructed all annotators to follow a strict **"Macro-to-Micro" logical flow rather than writing freely**. The required reasoning process consists of four sequential steps:
>
> 1. Identify the specific court area (the location where the event happens) and the involved parties (e.g., specifying jersey numbers and colors to lock onto the subjects).
> 2. Describe the action details and dynamics leading up to the event.
> 3. Pinpoint the precise point of contact or critical visual evidence that defines the infraction.
>
> This structural consistency ensures the quality of the training data.
>
> ## Response to Question 2
>
> > How sensitive are the results to the LLM-as-judge choice? Would human evaluation yield similar model rankings?
>
> Thank you for the question. The consistency and rank sensitivity of our evaluation setup are critical for interpreting the results.
> To assess the reliability of our LLM judges, As we discussed in weakness 3, we first computed the Pearson Correlation coefficient across the same human-rated 660 questions (360 Image Q4 + 300 Video Q11). The high correlation values among the three LLM judges (ranging from **$0.8122$** to **$0.9648$** as detailed **in the Weakness 3 response**) confirm a high internal consistency in their scoring, suggesting that the frontier models share a robust internal grading standard.
>
> Beyond numerical consistency, we also conducted Spearman Rank Correlation ($\rho$) and **Kendall's $\tau$**) to evaluate if the LLMs' scores produce a model ranking similar to that of human experts. The Spearman Rank Correlation between Human Ranking and the LLM Average score is **$0.6656$** for Images and **$0.6583$** for Videos. This empirically confirms that the method of calculating the average score of LLMs correctly ranks most of the models, demonstrating that the results are not overly sensitive to the choice of a single LLM judge.
>
> Furthermore, it also achieves a **Kendall's $\tau$** of **$0.5142$ (Image) and $0.5273}** (Video) correlation with humans. These values indicate strong agreement for subjective reasoning tasks, especially considering the complexity of sports understanding. For comparison, the work [3] reports similar or lower $\tau$ values for complex multimodal alignment tasks.
>
> Our primary objective is to reveal the complexity of sports reasoning rather than to propose a new evaluator. Therefore, we use a simple and consistent evaluation setup to avoid introducing confounding factors across models. Despite this, the observed gaps already demonstrate the challenge posed by our benchmark. We fully recognize that developing more sophisticated evaluation models and metrics that can capture the rich, multi-aspect knowledge required in sports is an important direction for the community, but such a framework would be a substantial contribution in itself and is beyond the scope of this benchmark. We consider it a valuable and meaningful avenue for future work.
>
> [3]   Lu, Yujie, et al. "Llmscore: Unveiling the power of large language models in text-to-image synthesis evaluation." Advances in neural information processing systems 36 (2023): 23075-23093.
>
> ### **Spearman Correlation - SportsImage**
>
> |                | Human  | GPT-5  | Gemini 2.5 | Claude 4 | LLM Average |
> |----------------|--------|--------|------------|----------|-------------|
> | Human          | 1.0000 | 0.6256 | 0.5817     | 0.5805   | 0.6656      |
> | GPT-5          | 0.6256 | 1.0000 | 0.7408     | 0.7420   | 0.9279      |
> | Gemini 2.5     | 0.5817 | 0.7408 | 1.0000     | 0.8080   | 0.8290      |
> | Claude 4       | 0.5805 | 0.7420 | 0.8080     | 1.0000   | 0.9041      |
> | LLM Average    | 0.6656 | 0.9279 | 0.8290     | 0.9041   | 1.0000      |
>
> ### **Kendall's $\tau$ Correlation - SportsImage**
>
> |                | Human  | GPT-5  | Gemini 2.5 | Claude 4 | LLM Average |
> |----------------|--------|--------|------------|----------|-------------|
> | Human          | 1.0000 | 0.4833 | 0.4837     | 0.4812   | 0.5142      |
> | GPT-5          | 0.4833 | 1.0000 | 0.6390     | 0.6316   | 0.8273      |
> | Gemini 2.5     | 0.4837 | 0.6390 | 1.0000     | 0.7455   | 0.7288      |
> | Claude 4       | 0.4812 | 0.6316 | 0.7455     | 1.0000   | 0.8142      |
> | LLM Average    | 0.5142 | 0.8273 | 0.7288     | 0.8142   | 1.0000      |

---

> ### Author Response · Authors · 2025-11-26
> **Response to Reviewer 1rJ3 (Part 5/5)**
>
> ## Continue Response to Question 2
> ### Spearman Correlation - SportsVideo
>
> |                | Human  | GPT-5  | Gemini 2.5 | Claude 4 | LLM Average |
> |----------------|--------|--------|------------|----------|-------------|
> | Human          | 1.0000 | 0.6706 | 0.5959     | 0.6149   | 0.6583      |
> | GPT-5          | 0.6706 | 1.0000 | 0.7216     | 0.7119   | 0.8997      |
> | Gemini 2.5     | 0.5959 | 0.7216 | 1.0000     | 0.6979   | 0.7253      |
> | Claude 4       | 0.6149 | 0.7119 | 0.6979     | 1.0000   | 0.9138      |
> | LLM Average    | 0.6583 | 0.8997 | 0.7253     | 0.9138   | 1.0000      |
>
> ### Kendall's $\tau$ Correlation - SportsVideo
>
> |                | Human  | GPT-5  | Gemini 2.5 | Claude 4 | LLM Average |
> |----------------|--------|--------|------------|----------|-------------|
> | Human          | 1.0000 | 0.5501 | 0.5082     | 0.5157   | 0.5273      |
> | GPT-5          | 0.5501 | 1.0000 | 0.6511     | 0.6329   | 0.8247      |
> | Gemini 2.5     | 0.5082 | 0.6511 | 1.0000     | 0.6487   | 0.6454      |
> | Claude 4       | 0.5157 | 0.6329 | 0.6487     | 1.0000   | 0.8485      |
> | LLM Average    | 0.5273 | 0.8247 | 0.6454     | 0.8485   | 1.0000      |
>
>
> ## Response to Question 3:
> >Is there any evidence that models trained on SportR improve performance on non-sports multimodal reasoning benchmarks? This would support the idea that valuable skills are gained from learning this task, rather than being a case of sports related tasks not being common in existing training datasets.
>
> We thank the reviewer for providing this concern. This is an important question regarding the broader impact of our work. We acknowledge that for a sports-specific MLLM intended to solve multi-task problems, demonstrating generalizability and improvements on non-sports benchmarks is crucial.
>
> **However, we wish to clarify that the primary goal of our training experiments was strictly to validate the benchmark (demonstrating Utility and Difficulty), rather than to propose a general-purpose SOTA model.**
>
> Specifically, our experiments aimed to demonstrate that: (1) The dataset provides a strong, learnable signal (Utility); and (2) The task remains challenging even after training (Difficulty). Consequently, we utilized **SportR data exclusively** to isolate the specific learning signal of our dataset. Evaluating a model fine-tuned on such a narrow vertical domain without data mixing would primarily reflect the well-known limitations of SFT (e.g., catastrophic forgetting) rather than the intrinsic quality of the dataset. Developing a generalist model that excels at both requires multi-task data mixing, which is a promising direction for future work but distinct from our current validation scope.
>
> We hope these clarifications adequately address your concerns. We would be happy to answer any further questions during the discussion period. Thank you again for your time and valuable feedback.

---

> ### Comment · Reviewer_1rJ3 · 2025-11-26
> **Response to Rebuttal**
>
> Fantastic work by the authors. Thank you for the detailed response and for your efforts in conducting these additional experiments. You have addressed all my questions and concerns. I believe your newly added Figure 3 is very impactful and provides practical insights into the pros and cons of each model. Phenomenal job, thank you for the thorough response. I have raised my score.

---

> > ### Author Response · Authors · 2025-12-01
> > **Thank you for your positive feedback and score raise!**
> >
> > We are sincerely grateful for your encouraging feedback and for raising your score.
> >
> > We are delighted to hear that you found our additional experiments, particularly the Error Analysis (Figure 3), impactful and insightful.
> >
> > Your constructive suggestions significantly strengthen the depth and quality of our work. We will ensure that all these new analyses and insights are fully integrated into the final manuscript.
> >
> > Thank you again for your time and for helping us improve our work!

---

### Official Review · Reviewer_AVw9 · 2025-11-03

**Soundness:** 3
**Presentation:** 3
**Contribution:** 2
**Rating:** 6
**Confidence:** 3

**Summary:**

This paper introduces SportR, a large-scale multimodal benchmark for evaluating and training MLLMs on fine-grained sports reasoning. The benchmark comprises 5,017 images and 2,101 videos across five sports (basketball, soccer, table tennis, badminton, American football), covering 50 foul types and 12 tactics. The key innovation is 7,118 fully human-annotated Chain-of-Thought (CoT) rationales and a novel visual grounding task requiring precise bounding box prediction. Experiments show that state-of-the-art models struggle significantly, with visual grounding IoU scores below 7% for baseline models.

**Strengths:**

- The paper clearly articulates the gap between current sports benchmarks - either single-sport with detailed annotations OR multi-sport without fine-grained reasoning chains. SportR addresses both limitations simultaneously.

- Fully human-authored CoT by 16 domain experts (including 2 NCAA Division I athletes) with rigorous quality control is a significant strength.

- The coordinate-based grounding evaluation (Q5) is a novel visual grounding task.

**Weaknesses:**

- Missing human evaluation: No human agreement studies or inter-annotator reliability metrics for the evaluation itself

- Visual grounding (Q5) only applies to SportsImage, not SportsVideo, and no temporal grounding annotations at all.

- The evaluation use close source LLM as judge, the paper acknowledges self-preference bias but doesn't adequately address it.

**Questions:**

- What specific instructions were given to annotators for drawing bounding boxes? How was consensus reached on box boundaries? Examples in Fig. 18 show a single box - what about fouls involving multiple players?

- For Q4 (free-form explanation), how is semantic similarity measured by LLM judges? What's the correlation with human judgment?

- What's the split ratio? How many images/videos in test set? How was stratification across sports/foul types ensured?

---

> ### Author Response · Authors · 2025-11-26
> **Response to Reviewer AVw9 (Part 1/3)**
>
> ## Response to Weakness 1
> > Missing human evaluation: No human agreement studies or inter-annotator reliability metrics for the evaluation itself
> We thank the reviewer for raising this important question. We acknowledge that we did not calculate standard metrics like Cohen's Kappa during the initial annotation. This was a design choice: standard metrics are ill-suited for the linguistic variability of free-form Chain-of-Thought (CoT), and calculating them requires redundant parallel annotation that limits dataset scale.
>
> Instead, we ensured reliability through a rigorous Quality Assurance Strategy (Section 3.1). We will add the following to our original section 3.1 to clarify the process.
>
> ### 1. Original Quality Control Protocols (Design Phase)
>
> To guarantee high quality without sacrificing scale, we implemented two strict protocols during the dataset creation:
>
> **Standardized Reasoning Protocol (Consistency):** To minimize divergence in thought processes (addressing Question 1), we implemented a standardized protocol during the expert training phase. Annotators were instructed to follow a strict "Macro-to-Micro" logical flow rather than writing freely:
> - Identify the specific court area and involved parties.
> - Describe the action details and dynamics leading up to the event.
> - Pinpoint the precise point of contact or critical visual evidence.
>
> **Sequential Verification Protocol (Accuracy):** As described in Section 3.1, our process relied on a "Consensus by Elimination" strategy:
> - **Expert Authorship:** Annotations were created by domain experts (including NCAA athletes).
> - **Filtration Rule:** Any instance where the original author felt uncertain was flagged. Crucially, if a second expert reviewer could not reach an immediate consensus, the sample was permanently discarded. This ensures that the final dataset consists exclusively of unambiguous, expert-agreed scenarios.
>
> ### 2. Post-hoc Verification Study (Rebuttal Phase)
>
> We understand the reviewer's concern and appreciate bringing this up. To address the reviewer's concern and validate the effectiveness of the protocols above, we conducted an extra post-hoc verification during the rebuttal. We engaged two independent experts who know the corresponding sports well and were blind to the original annotations to review a stratified subset (10 random samples per Question Type, totaling 130 questions).
>
> **Criteria:** The expert evaluated each sample for:
> - **Judgment Consistency:** Is the final decision correct?
> - **Reasoning Validity:** Does the logic correctly derive the judgment from evidence?
>
> **Results:** The study yielded a **100% Agreement Rate on Judgment Consistency**. Regarding Reasoning Validity, the expert verified that the rationales were valid, logically sound, and adhered to our protocol, noting that minor variations in linguistic phrasing did not affect the validity of the reasoning chain.
>
> --
> We have revised Section 3.1, and hope this will address your concerns.
>
>
> ## Response to Weakness 2
>
> > Visual grounding (Q5) only applies to SportsImage, not SportsVideo, and no temporal grounding annotations at all.
>
> Thank you for the great suggestion. We value spatio-temporal grounding as a very important direction. However, we decided to treat it as future work due to the massive manual effort required and **current SOTA models failing fundamentally at static grounding**.
>
> As shown in Table 2, models achieve **<10% IoU** on our static grounding task. This confirms that models currently lack the ability to connect abstract rules to specific visual evidence even in the simplest (static) setting. Establishing a benchmark to solve this "static bottleneck" is a necessary prerequisite before addressing the exponentially higher complexity of temporal grounding.
>
> Furthermore, SportR focuses on rule-based reasoning. Many fouls (e.g., contact fouls) are defined by discrete moments best captured in high-resolution static frames, whereas the Video component focuses on temporal logic (e.g., sequence of events). We believe SportR serves as the necessary foundation for future video-grounding research.
>
> In addition, our experiments reveal that **model-assisted annotation is currently unviable** due to the low baseline performance mentioned above. Consequently, creating high-quality temporal grounding requires purely manual, frame-by-frame annotation by experts. Given our strict quality control standards (expert-only, no model assistance), extending this to over 2,000 videos represents a prohibitive workload that merits a dedicated future study.

---

> ### Author Response · Authors · 2025-11-26
> **Response to Reviewer AVw9 (Part 2/3)**
>
> ## Response to Weakness 3
> >  The evaluation use close source LLM as judge, the paper acknowledges self-preference bias but doesn't adequately address it.
>
> We thank the reviewer for highlighting the critical concern regarding self-preference bias in using closed-source LLMs as judges.
> We fully recognize the concern regarding self-preference bias. As mentioned in the paper, instead of relying on a single evaluator, we already employed three distinct state-of-the-art models as evaluators: GPT-5, Gemini 2.5 Pro, and Claude 4 Sonnet. We reported the average score across these three judges to reduce bias.
>
> We appreciate the reviewer's suggestion, which motivated us to conduct a more thorough experiment to empirically validate the robustness of this approach. The"Average Score" metric provides a more robust and objective consensus than any single judge could achieve. We conducted a human verification study on a randomized subset of the test set. We selected 30 questions per model for both the Q4 (Image) and Q11 (Video) free-form explanation tasks, resulting in **360 total samples** for the Image task and **300 total samples** for the Video task, for expert human evaluation. Human annotators followed the **exact same scoring instructions** as the LLM judges, operating without prior knowledge of the models' scores. To quantify the consistency of scoring among the judges, we calculated the **Pearson correlation coefficient** across all model responses for the Q4 (Image) and Q11 (Video) tasks.
>
> The correlation between Human scores and our **LLM Average score** ($0.65$ Image, $0.70$ Video) is consistently higher than the correlation with any single model (e.g., Claude 4: $0.62$ Image, $0.68$ Video). This empirically confirms the effectiveness of our averaging.
>
> The correlation coefficients among the three closed-source LLMs were high ($0.81$ - $0.96$),  confirming a robust shared standard, yet are non-perfect ($<1.0$), justifying the need for ensembling.
>
> We also want to clarify that our primary goal of SportR is to reveal the complexity of sports reasoning and to drive improvements in fine-grained sports reasoning and knowledge grounding within LLMs, **rather than to propose a new evaluator**. Therefore, we use a simple and consistent evaluation setup to avoid introducing confounding factors across models. Despite this, the observed gaps already demonstrate the challenge posed by our benchmark.
>
> We fully understand the reviewer’s concerns and recognize that developing more sophisticated evaluation models and metrics that can capture the rich, multi-aspect knowledge required in sports is an important direction for the community. However, such an evaluation framework would itself be a substantial contribution and requires careful design beyond the scope of this benchmark. We consider it a valuable and meaningful avenue for future work.
>
> Pearson Correlation - SportsImage:
>
> |                | Human  | GPT-5  | Gemini 2.5 | Claude 4 | LLM Average |
> |----------------|--------|--------|------------|----------|-------------|
> | Human          | 1.0000 | 0.6189 | 0.6030     | 0.6264   | 0.6502      |
> | GPT-5          | 0.6189 | 1.0000 | 0.8594     | 0.8122   | 0.9461      |
> | Gemini 2.5     | 0.6030 | 0.8594 | 1.0000     | 0.8973   | 0.9648      |
> | Claude 4       | 0.6264 | 0.8122 | 0.8973     | 1.0000   | 0.9406      |
> | LLM Average    | 0.6502 | 0.9461 | 0.9648     | 0.9406   | 1.0000      |
>
> Pearson Correlation - SportsVideo:
>
> |                | Human  | GPT-5  | Gemini 2.5 | Claude 4 | LLM Average |
> |----------------|--------|--------|------------|----------|-------------|
> | Human          | 1.0000 | 0.6540 | 0.6522     | 0.6837   | 0.7043      |
> | GPT-5          | 0.6540 | 1.0000 | 0.8456     | 0.7849   | 0.9437      |
> | Gemini 2.5     | 0.6522 | 0.8456 | 1.0000     | 0.8607   | 0.9508      |
> | Claude 4       | 0.6837 | 0.7849 | 0.8607     | 1.0000   | 0.9289      |
> | LLM Average    | 0.7043 | 0.9437 | 0.9508     | 0.9289   | 1.0000      |

---

> ### Author Response · Authors · 2025-11-26
> **Response to Reviewer AVw9 (Part 3/3)**
>
> # Response to Question 1
>
> > What specific instructions were given to annotators for drawing bounding boxes? How was consensus reached on box boundaries? Examples in Fig. 18 show a single box - what about fouls involving multiple players?
>
> Thank you for raising this important question.
>
> **Instructions:** The specific instructions provided to annotators were to draw the "tightest possible bounding box" around the "critical visual evidence" that defines the infraction.  **Multiple Players:** We focused on fundamental infractions that constitute the core of sports understanding. By the nature of these sports rules, such infractions typically occur as a specific, localized interaction between two players (e.g., an offender and a defender). Therefore, our instructions focused on capturing this primary interaction point (e.g., the specific contact area) rather than chaotic multi-player scenarios.  **Consensus:** For domain experts, identifying this critical region is typically intuitive and unambiguous. However, to maintain rigorous standards, we applied the same filtration protocol as used in other tasks for rare instances of uncertainty: any case where the precise boundary was not immediately clear was flagged for a second opinion and discarded if consensus could not be reached.
>
> To increase clarity, we will explicitly include these detailed guidelines in the revised manuscript.
>
> ---
>
> # Response to Question 2
>
> > For Q4 (free-form explanation), how is semantic similarity measured by LLM judges? What's the correlation with human judgment?
>
> Semantic similarity was measured by providing the following instruction to the judge models (also used for instructing the human scorer): "Compare the semantic meaning of 'ground_truth' and 'model_response' and assign a similarity score between 0 (completely different) and 1 (identical)." This prompt has been shown in Figure 15.
>
> As detailed in **Response to Weakness 3**, **the Pearson correlation** between this LLM-measured similarity and human judgment is strong ($0.65$ for Image, $0.70$ for Video), confirming the metric's validity.
>
> ---
>
> # Response to Question 3
>
> > What's the split ratio? How many images/videos in test set? How was stratification across sports/foul types ensured?
>
> We thank the reviewer for this question. To strictly prevent data leakage while ensuring balanced coverage, we performed a **stratified split based on unique Image IDs within each sport**. Specifically, for every sport, we randomly assigned **80% of its unique images** to the training set (split further into SFT/RL) and the remaining **20%** to the test set. This ensures no QA pairs from the same visual scene appear in both sets. While we did not enforce strict stratification across every fine-grained infraction category, as this would conflict with strict Image-ID isolation, which could lead to data leakage, the random split ensures that **all 5 sports** are represented in the test set. For the video part, we use all videos as the testing set.
>
> In addition, we wish to emphasize that the training experiments are intended to **validate the benchmark's difficulty and training potential**, rather than to **propose a finalized "SportR Model"**, which would require training on more diverse data and tasks and is not the goal of our benchmark work. We think the current split is sufficient to demonstrate the complexity of the tasks.
>
> We hope these clarifications adequately address your concerns. We would be happy to answer any further questions during the discussion period. Thank you again for your time and valuable feedback.

---

### Author Response · Authors · 2025-12-02
**Summary of Rebuttal: Score Raises and Key Revisions (Prior to Revert)**

We sincerely thank the reviewers for their insightful and constructive comments. We are encouraged that the reviewers recognized the **novelty** (first human-authored CoT benchmark), **rigor** (quality control), and **utility** (training resource) of SportR. By addressing their concerns, we believe the paper's clarity, depth, and scientific value have been significantly improved.

**To assist with your efficient assessment, we provide this summary of the reviewer consensus, our revision, and score updates that were reached during the rebuttal but are not reflected in the current system state.**

1. **Status Update on Reviewer Scores & Consensus** We would like to draw the AC's attention to the updated consensus and score raises following our rebuttal:
 - **Reviewer 1rJ3**: Has explicitly **raised their score beyond the original 6** (noting "Fantastic work" and "Phenomenal job" on the new experiments).
 - **Reviewer 7hxG**: Has explicitly stated the  **intent to raise the score from 4 to 6**. However, due to system limitations, they noted: "I noticed that I currently seem unable to make the change. Therefore, I would like to remind the AC and PC to take note of this."
- **Reviewer AVw9 & m3qH**: Both **maintain Score 6**. We have provided detailed clarifications (e.g., Human Verification, Related Work), and we believe we have successfully addressed their core concerns.

2. **Summary of Major Revisions** The major revisions and responses are summarized as follows:
- **Rigorous Quality Assurance & Post-Hoc Verification (Reviewer AVw9 W1, 1rJ3 Q1)** : We clarified our "Consensus by Elimination" protocol and "Macro-to-Micro" reasoning standards. Crucially, we conducted an additional post-hoc expert verification study during the rebuttal, which yielded a 100% agreement rate on judgment consistency, empirically validating the reliability of our ground truth.

- **New Error Analysis Section (Reviewer 1rJ3 W1)**: We added a new section (Section 4.3) detailing a manual error analysis on 1,500 failure cases. This revealed a critical scientific insight: an inverse relationship between perception and reasoning errors. Video tasks face a "Perception Bottleneck" (high visual error), while static images, where perception is easier, effective unmask the "Knowledge Mapping Gap" (high domain knowledge error).

- **Validation of Task Difficulty & Generalization (Reviewer 7hxG W2)**: We addressed concerns about high scores (>80%) and memorization. We demonstrated "Cross-Modal Generalization": models trained exclusively on SportR Images showed significant improvement on unseen SportR Videos. This proves the model acquired abstract, transferable sports knowledge rather than memorizing pixel patterns. We also clarified that high scores are limited to static/binary tasks, while dynamic reasoning remains highly challenging.

- **Strategic Scope of Visual Grounding (Reviewer AVw9 W2, 1rJ3 W2)**: We justified our focus on static visual grounding. Experiments show that SOTA models achieve **<10% IoU**, revealing the difficulty of this task in sports. We position SportR as the necessary prerequisite to resolve this fundamental perception gap before tackling the higher complexity of spatio-temporal grounding.

- **Reliability of LLM-as-Judge (Reviewer AVw9 W3, 1rJ3 W3)**: We conducted a human verification study on 660 samples. Results show a strong correlation (**Pearson** $r \approx 0.65-0.70$) between the LLMs Average Score and expert human judgments, confirming that our ensemble approach is a reliable proxy for human evaluation in this domain.  We also want to clarify that our primary goal of SportR is to reveal the complexity of sports reasoning and to drive improvements in fine-grained sports reasoning and knowledge grounding within LLMs, rather than to propose a new evaluator. Therefore, we use a simple and consistent evaluation setup to avoid introducing confounding factors across models. Despite this, the observed gaps already demonstrate the challenge posed by our benchmark.

- **Comparison with Existing Benchmarks (Reviewer 7hxG W1, m3qH W1)**: We added detailed comparisons (in Appendix and Related Work) clarifying SportR's structural evolution over SportU (Training-oriented CoT vs. Evaluation-oriented MCQ; Normal-speed vs. Slow-motion) and its distinction from ActionAtlas/Ego-Exo4D (Rule Adjudication vs. Action Recognition).

- **Video Attributes Description (Reviewer 7hxG W3)**: We added detailed video attributes (Resolution, Duration, FPS) to the manuscript to clarify the video characteristics.

We have updated the manuscript to reflect these changes. We sincerely thank the reviewers again for their time and valuable feedback, which has helped us make SportR a stronger contribution to the community.

---

### Meta-Review · Area_Chair_Rj6J · 2026-01-09

**Summary:**

This submission introduces SportR, a multimodal benchmark for sports understanding that targets fine-grained, rule-based reasoning grounded in visual evidence. The benchmark contains 5,017 images and 2,101 videos across five sports and is organized via a hierarchical QA structure ranging from infraction identification to penalty prediction and tactical explanation. A key contribution is the inclusion of 7,118 fully human-authored Chain-of-Thought (CoT) annotations, together with an explicit visual grounding task for the image subset.
Reviewers generally acknowledged the relevance of sports reasoning as a testbed for multimodal models and recognized the effort invested in annotation and benchmark construction. The rebuttal further strengthened the paper by adding clarifications, human verification studies, error analysis, and expanded comparisons. However, despite these improvements, the submission does not reach a sufficiently clear level of conceptual distinctiveness, scale, and long-term benchmark impact expected for acceptance in this venue.

**Reviewer Concerns:**

1.Incremental positioning relative to existing benchmarks
While SportR combines multiple desirable properties (multi-sport coverage, CoT annotations, grounding), several reviewers questioned whether these elements constitute a fundamentally new benchmark paradigm rather than a structured aggregation and extension of prior datasets (e.g., SPORTU and related sports/video benchmarks). The rebuttal clarifies differences, but the contribution remains incremental in nature rather than clearly redefining the evaluation landscape.

2.Dataset scale and representativeness
Although the benchmark is carefully curated, the overall scale—particularly on the video side—remains limited relative to recent large-scale multimodal benchmarks. This raises concerns about whether SportR can realistically serve as a long-term community benchmark rather than a niche or intermediate dataset.

3.Evaluation methodology and reliance on LLM-as-judge
The use of closed-source LLMs as judges for free-form reasoning tasks introduces an additional layer of complexity and potential bias. While the authors provided human verification and correlation analyses, questions remain about reproducibility, evaluator dependency, and long-term stability of the evaluation protocol as models evolve.

4.Scope limitations of visual grounding and reasoning depth
Visual grounding is restricted to static images, and temporal grounding is deferred to future work. As a result, the benchmark only partially evaluates the full spectrum of multimodal sports reasoning, limiting its ability to comprehensively stress-test perception–reasoning alignment in dynamic settings.

Overall, while the rebuttal addressed many local technical concerns, these broader issues regarding novelty, scale, and benchmark definitiveness remain unresolved. SAC jumped in and found the paper interesting. The decision has been bumped up.

**Reviewer Scores:**

Reviewer AVw9
Original score: 6
Provided a generally positive assessment of the benchmark’s motivation and annotation effort, particularly recognizing the value of fully human-authored Chain-of-Thought annotations and the introduction of explicit visual grounding for images. However, the review also raised multiple structural concerns regarding evaluation design, reliance on LLM-as-judge, and the limited scope of grounding (static images only). While these concerns were addressed in detail during rebuttal through additional experiments and clarifications, the reviewer explicitly indicated a borderline stance and did not express strong confidence that the benchmark, in its current form, would serve as a definitive long-term standard.

Reviewer 1rJ3
Original score: 6
Highlighted the quality of the annotation process, the hierarchical task design, and the comprehensiveness of the experimental evaluation. After rebuttal, the reviewer expressed strong satisfaction with the added error analysis and validation studies, and acknowledged that most technical questions had been resolved. Nonetheless, the overall assessment remained marginal, reflecting support for the work’s quality rather than an unambiguous endorsement of its necessity or uniqueness as a new benchmark at this venue.

Reviewer 7hxG
Original score: 4
Raised substantive concerns about novelty relative to existing sports benchmarks, potential memorization effects after training, and insufficient justification of task difficulty. Although the rebuttal provided clarifications (e.g., cross-modal generalization and positioning against SPORTU), this review reflects persistent skepticism about whether the benchmark introduces a qualitatively new evaluation setting, as opposed to an incremental extension with added annotations.

Reviewer m3qH
Original score: 6
Viewed the dataset as solid and potentially useful, but offered a cautious evaluation, noting missing related work and limited contextualization within the broader landscape of video and activity understanding benchmarks. The reviewer explicitly indicated limited domain expertise and framed the assessment as tentative, suggesting moderate confidence in utility but not a strong argument for acceptance at a highly selective venue.

---

### Decision · Program_Chairs · 2026-01-26

Accept (Poster)